# Constructing HIV/AIDS in Chinese media (2010–2024): A mixed-methods study

Yuhang Li[1], Chenghui Wu[2], Lisai Yu[3], Haiyong Shi[4]*

**1** Zhijiang College, Zhejiang University of Technology, Shaoxing, Zhejiang, China, **2** School of Foreign Languages, Peking University, Beijing, China, **3** Center for the Study of Language and Cognition, Zhejiang University, Hangzhou, Zhejiang, China, **4** Department of Public Health, Lishui People's Hospital, Lishui, Zhejiang, China

* haiyongshi@outlook.com

## Abstract

In China, where HIV/AIDS remains one of the leading causes of infectious disease-related mortality, traditional media significantly shape public perceptions amid persistent prevention challenges. This study examines a large-scale corpus of news articles published between 2010 and 2024 to understand the evolving media discourse surrounding HIV/AIDS. Employing Analysis of Topic Model Networks (ANTMN) and collocation analysis, we extract the thematic networks, terminology for people living with HIV (PLHIV), and HIV/AIDS metaphors. Multiple Correspondence Analysis (MCA) and network backbone method were complementarily employed to examine the interrelationships among discursive strategies and their connections with thematic contexts. Our analysis identifies five major thematic communities: "Prevention and control", "Publicity", "Society", "Medicine", and "PLHIV", encompassing 48 distinct topics. These findings reveal a clear discursive shift towards political and macro-level narratives, moving away from individual and social perspectives. Collocation analysis reveals 19 categories of PLHIV terminology and 12 categories of HIV/AIDS metaphors. Although de-identified terminology has become mainstream in discussions about PLHIV, stigmatizing terminology persists with typological diversity. War, journey, and entity metaphors form the core conceptual framework. The associations among PLHIV terminology, metaphors, and thematic contexts reflect the strategic adaptations of media institutions within a state-dominated system, while simultaneously manifesting the entrenched inertia of stigmatization. This study updates the empirical landscape of HIV/AIDS discourse within China's media context, offering new insights into how the media shape social cognition of HIV/AIDS.

## Introduction

The Joint United Nations Programme on HIV/AIDS (UNAIDS) has set the ambitious goal of eliminating HIV/AIDS as a public health threat by 2030. However, achieving

**Data availability statement:** All relevant data are within the paper and its Supporting information files. The code used in this study is available in the Figshare repository: https://doi.org/10.6084/m9.figshare.30985558.

**Funding:** The author(s) received no specific funding for this work.

**Competing interests:** The authors have declared that no competing interests exist.

this goal remains a formidable challenge, especially in China. According to the Chinese Center for Disease Control and Prevention (China CDC), HIV/AIDS has become a leading cause of death from infectious diseases [1], with the continuously increasing number of newly identified PLHIV infections, diagnostic delays, and associated HIV/AIDS risks among key populations, such as homosexual men [2]. Amid this severe prevention and control landscape, mass media serves as a primary conduit for public health information, wielding substantial influence over societal perceptions and behavioral responses to HIV/AIDS. Yet, media representations are rarely neutral; their narrative frameworks, discursive strategies, and metaphorical structures can subtly shape public understanding, potentially exacerbating social prejudice and discrimination against people living with HIV [3].

As a powerful instrument for shaping public opinion [4], mass media play a dual role in the HIV/AIDS discourse [5]. On one hand, news media function as a vital channel for public health education, enhancing awareness, disseminating prevention strategies, and promoting behavioral change [5–7]. Authorities, particularly in developing nations, also leverage the media to mitigate HIV/AIDS-related stigma [8]. On the other hand, the media act as a dominant force in constructing HIV/AIDS discourses, where negative narratives and sensational language targeting PLHIV can reinforce stereotypes and perpetuate discrimination [3,9,10]. These discourses profoundly reflect underlying social structures, cultural contexts, and power mechanisms [11]. For instance, top-down power operations may foster bureaucratic tendencies that stigmatize marginalized groups [12], while media framing exhibits distinct characteristics under varying political systems or ideologies [13]. The media simultaneously reflect social structures and amplify public sentiment. They often establish binary "us versus them" narratives through excessive associations between the disease and specific "high-risk groups" or entrenching stereotypes via disproportionate reporting of certain social groups [14,15]. Crucially, HIV/AIDS media discourse represents an evolving process, dynamically adapting to shifting societal landscapes over time [16–18].

Scholars have identified various discursive strategies employed in HIV/AIDS coverage, including modal assessment to reinforce social norms [19], predicative strategies like verb choice or passive voice to construct power relations [20,21], and quotations for intertextual analysis [22]. Among these, referential strategies and metaphors are particularly crucial. Referential strategies, operating through intra-group and inter-group divisions, determine how the media define HIV/AIDS and associated populations [23]. This occurs via mechanisms such as categorization (e.g., constructing stigmatizing dichotomies like "evil versus victim" in criminal prosecution coverage, portraying PLHIV as morally deficient) [10,24] and de-spatialization (e.g., confining the epidemic to specific regions, creating boundaries between safety/danger or purity/pollution) [25]. Metaphors, reconceptualized by cognitive linguistics as fundamental cognitive mechanisms [26], shape public cognition through mappings from concrete source domains to abstract target domains. Sontag [27] identified dual genealogies in disease metaphors: "invasion" for progression and "contamination" for

transmission. Through such rhetoric, diseases are transformed from medical concerns into public issues entangled with morality, politics, and ideology [28–30].

While media representations of HIV/AIDS have been extensively studied globally [11,31], significant limitations persist. Most studies are constrained by small sample sizes, hindering the capture of dynamic evolution amid social environments shifts and policy changes. This is particularly evident in the scarcity of systematic analyses focusing on Chinese media coverage over the past decade and beyond. Early research on HIV/AIDS coverage in China predominantly employed qualitative methodologies (e.g., interviews, close textual reading) [5,32,33] or small-scale content analyses [34,35], which, while enabling in-depth exploration, lack representativeness for comprehensive overviews. Although there exist diachronic studies [3,13,17], they primarily focus on the first decade of the 21st century or, more recently, on social media [11], leaving a critical gap in understanding how traditional mainstream media have constructed HIV/AIDS in recent years amid societal and medical advancements. Although social media have fragmented audiences and reduced traditional readership, a focus on mainstream media, especially party and commercial newspapers, remains methodologically and contextually critical. First, traditional outlets retain unparalleled institutional authority, acting as official barometers for state public health policy [36]. Second, they function as foundational agenda-setters, providing the authoritative discourse templates that are subsequently amplified on social media [37]. Finally, for a 15-year longitudinal study, traditional media offer a stable, institutionally archived textual record [38], avoiding the ephemeral nature and algorithmic biases inherent in social media data. Furthermore, existing research has predominantly focused on thematic and framing analysis [24,39] which reveals macro-level topic distributions but often fails to capture nuanced discursive strategies embedded within the discourse, particularly the linguistic devices like lexical choices and metaphors that shape public perceptions.

Against this backdrop, we employ newspapers as a proxy indicator for epidemic-related media representation [40,41]. Drawing upon a large-scale textual corpus of HIV/AIDS coverage from Chinese traditional media from 2010 to 2024, this study analyzes the construction of HIV/AIDS in media discourse and its temporal evolution. Particular attention is paid to the terms used to refer to PLHIV and the metaphors related to HIV/AIDS. The research addresses three key questions: 1) What structural patterns and evolutionary characteristics have emerged in the thematic focus of media coverage over the past 15 years? 2) What PLHIV terminology and HIV/AIDS metaphors are employed by the media to construct public perceptions? 3) What are the relationships among different types of discourse strategies, as well as between discourse strategies and thematic contexts?

This large-scale, longitudinal study not only provides an empirical foundation for understanding the local trajectory of media discourse but also enriches the interpretation of HIV/AIDS and social cognition in the Chinese context, offering critical insights for public health practices and policy optimization aimed at achieving the 2030 elimination goal.

## Methodology

### Data collection

We collected data extensively through online retrieval platforms and publicly accessible databases of news publishing houses. The target media were restricted to officially circulated traditional media in mainland China, encompassing authoritative outlets such as People's Daily and China News Network, as well as regional and specialized media. Given that headlines perform the function of guiding readers to understand the nature of the full text [42], we used *aizi* (HIV/AIDS) and "HIV" as search terms. Data were collected from the news websites between June 2 and July 1, 2025. Covering the period from January 1, 2010, to December 31, 2024, we collected all news articles whose headlines contained these search terms, and then conducted our analysis on the full texts of these articles. This approach ensured strong relevance to HIV/AIDS topics while minimizing the influence of irrelevant texts.

The selection of 2010 as the starting point for this research is primarily driven by the following considerations. Existing studies on HIV/AIDS reporting in Chinese mainstream media have mostly concentrated on the period from the 1990s to

2010 [3,13,17], while a systematic investigation into the evolution of coverage after 2010 remains scarce. Simultaneously, the year 2010 signifies a threefold critical turning point in China's public health context. Epidemiologically, sexual transmission replaced intravenous drug use and blood-borne transmission as the primary infection route in that year, which notably has broadened the social impact of the disease [43]. In terms of policy, China abolished its entry restrictions for HIV-positive foreigners in April 2010, marking a vital transition in public health governance [44]. Meanwhile, the rapid emergence of social media around 2010 fundamentally reshaped China's media landscape, forcing traditional media to persistently adjust their discursive strategies [45]. Consequently, examining HIV/AIDS reporting over the 15 years from 2010 to 2024 is of significant academic and practical importance. In total, we gathered 40,038 raw news articles, which sufficiently represented the primary body of HIV/AIDS coverage in Chinese media during this period.

**Topic network**

We selected the recently developed Analysis of Topic Model Networks (ANTMN) as the tool for topic exploration. This method addresses the limitations of manual coding and remedies the deficiencies of traditional Latent Dirichlet Allocation (LDA) approaches [46] by innovatively integrating unsupervised machine learning with network analysis [47]. It combines latent topic modeling and community detection algorithms to extract topics as frame elements, then constructs co-occurrence networks that identify discourse content as communities, with distinct content demarcated by community boundaries. For a more detailed description of the methodology and comparisons with existing alternatives, we referred readers to the original publication. In summary, we adhered to the procedural steps outlined by its authors [47].

The first step is topic modeling. We began by preprocessing the corpus, which involved an initial cleaning stage comprising the following steps: (1) unifying letter cases; (2) Traditional-to-Simplified Chinese conversion; (3) duplicate removal; and (4) symbol elimination. Additionally, following Walter and Ophir's [48]recommendations, we excluded extremely short news articles (<50 characters), non-Chinese articles, and excessively long policy documents. Subsequently, we conducted word segmentation using the jieba Chinese word segmentation tool, applying both common stopword lists and domain-specific lexicons. Concurrently, adhering to standard procedures [49], we retained terms with frequency > 3 and document frequency < 99.00%. Based on Chinese linguistic characteristics, we also filtered out non-numeric single-character terms. We then employed LDA to evaluate latent topics across all news articles. During this process, we further removed documents that became blank after cleaning and filtering, as these contained no information useful for topic analysis and were deemed invalid. Ultimately, we obtained 39,141 valid documents processed through topic modeling, with each document assigned probability scores for all topics by the algorithm.

To determine the optimal number of topics, we employed a ten-fold cross-validation, iterating through a series of topic quantities (from k = 10 to k = 150 at intervals of 10). We calculated coherence scores for models with each k value, determining the optimal topic number by synthesizing the Elbow Method and perplexity metrics. Through this exhaustive process, we identified the model with k = 50 as optimal. Subsequently, following the standard procedure for assigning topic labels [47], we qualitatively labeled each topic by examining its top terms, exclusive terms, and representative documents. Topics with similar semantic content were distinguished by sequential numbering under the same label. During this process, we excluded "boilerplate" topics, which lacked substantive meaning [49]. Ultimately, we obtained 48 topics with clearly defined labels. All subsequent analyses were conducted based on these 48 topics to ensure statistical consistency.

In the second step, we constructed a topic community network. Based on the document-topic matrix, we generated a similarity matrix using cosine similarity to quantify the strength of node associations. Treating these similarities as edges, we applied the backbone extraction method [50] to build a network where each topic served as a node and their co-occurrences as edges. This method evaluates edge significance through statistical testing, determining whether each

edge is more important than would occur randomly. By retaining edges with p-values below a threshold, we effectively eliminated spurious connections. For the backbone threshold, we followed the approach of Ophir et al. [51] and selected the critical point where the network began to fragment (p < .37 in our data). This choice prioritized maintaining a connected network—essential for community detection in topic networks derived from news texts.

In the third step, we employed the Louvain community detection algorithm [52] to group topics into broader thematic communities. This algorithm is recommended in related studies [47,51] for its effectiveness. To quantify the share of language associated with each community in individual news articles, we summed the probabilities for all topics linked to the specific community within each article.

## Collocation extraction

Lexical collocations are co-occurrences of words formed according to semantic rules. Through lexical collocations, we can observe the construction strategies of news media regarding PLHIV and the disease itself, as these collocations reflect the inherent perceptions of news producers. In particular, specific terminology collocations referring to PLHIV often contain media stereotypes about certain groups [3]. Meanwhile, metaphorical collocations about HIV/AIDS can reflect more hidden cognitive models [53]. Previous relevant studies have often been limited by scale and tools, only able to analyze a limited range of PLHIV terminology or metaphors [3].

In this study, we leverage natural language processing to discover more terminology for PLHIV and HIV/AIDS metaphors. Traditional collocation extraction work often relies on classic corpus software such as Antconc or Wordsmith Tools. These tools are suitable for smaller corpus sizes and are not highly efficient. We adopt a method based on Pointwise Mutual Information (PMI), which measures the ratio of co-occurrence probability and independent occurrence probability of two words, using the gensim package in Python 3.4 to directly extract all possible collocations from the news corpus. This model can adjust the threshold to change the degree of fixedness of the extracted collocations. Since PLHIV terminology and HIV/AIDS metaphors are not strictly fixed collocations but rather word combinations that co-occur relatively easily compared to free combinations, we set a lower threshold (threshold = 0.1), obtaining a total of 1,158,316 potential collocations.

Next, we utilized regular expressions to extract all collocations containing HIV/AIDS related expressions (totaling 4,424). These collocations represent fixed word combinations most likely to contain PLHIV terminology and HIV/AIDS metaphors, while minimizing the impact of irrelevant collocations. Subsequently, two trained coders (graduate research assistants with expertise in health communication) independently annotated all 4,424 HIV/AIDS-related collocations extracted from the corpus. To avoid bias from predefined categories, coders freely generated unique category labels for each collocation based on semantic emphasis, extracting potential PLHIV terminology and HIV/AIDS metaphors. Drawing on the existing HIV/AIDS stigma research [3,8] and established metaphor list [54], the first author classified labels with different expressions but the same meaning. Following this, considering that some categories contained fewer collocations, we employed Krippendorff's Alpha to examine inter-coder reliability [55]. The results showed high consistency, with the value of 0.83 for terminology annotations and 0.81 for metaphor annotations, demonstrating the reliability of the lexicon. Finally, only collocations achieving consensus through discussion among all experts were retained. A total of 19 terminological categories pertaining to PLHIV were identified, encompassing 148 collocations. These included "Generic" terminology without specific markers such as gender or age, alongside stigmatizing terminology targeting particular populations, which were classified based on biological attributes, roles, group affiliations, and value judgments. Additionally, 12 categories of HIV/AIDS metaphors were identified, comprising 170 collocations.

## Co-occurrence pattern

We employed co-occurrence data to examine the interrelationships among different discursive strategies, as well as their connections with thematic contexts. Firstly, we conducted multiple correspondence analysis (MCA) involving PLHIV

terminology and HIV/AIDS metaphors from different themes. Secondly, a Jaccard matrix was constructed based on distributional data of discursive strategy, and we similarly employed Serrano's [50] backbone method to identify significant links within this matrix. This backbone method is adaptable to various scenarios depending on the selected threshold. A standard p-value (p < .05) was chosen as the threshold for discovering significant co-occurrence patterns. Under stricter thresholds, the network may decompose into multiple components. The components remaining connected at this threshold represent significant co-occurrence patterns, while discursive strategies not connected together remain relatively independent.

To further investigate the significant co-occurrence patterns between terminology designating specific populations and particular HIV/AIDS metaphors, we conducted a targeted qualitative analysis on the intersecting articles. Drawing upon the cognitive metaphor analysis framework established by Lakoff and Johnson [26] and further applied in critical health discourse, two trained researchers (Ph.D. candidates with backgrounds in linguistics) independently performed manual coding on all news reports containing the metaphors (the specific counts are reported in the Results section) to identify the underlying source domain elements of these metaphors. For the coding rules of this process, we followed the identification guidelines of the Metaphor Identification Procedure Vrije Universiteit (MIPVU) alongside established practices in cognitive metaphor research [56,57]. Specifically, taking entity metaphors as an example, a coder labeled the entity metaphor when encountering Chinese expressions in a news report where HIV/AIDS was conceptualized as a discrete entity or substance (e.g., the term *shoulian*, meaning "convergence"). Subsequently, based on the semantic properties of the vehicle terms, the metaphor was assigned to specific source domain elements (e.g., controllability, boundaries, etc.) [58]. The initial inter-coder reliability reached a satisfactory level (Krippendorff's Alpha = 0.77), with all remaining differences resolved through discussion to achieve final agreement.

## Results

### Themes

As shown in Fig 1, our model identified 48 topics, which were clustered into 5 community networks representing 5 major themes.

**Theme 1.** The first community network focuses on macro-level prevention and control, predominantly associated with government and policies. We have designated this the "Prevention and control" theme, formed by nine interconnected topics (blue nodes) in Fig 1. This thematic community collectively presents a comprehensive, highly policy-driven HIV/AIDS prevention and control system. Media coverage within this theme emphasizes national prevention strategies (e.g., "Prevention and treatment", "Policy", "Prevention" topics), continuously tracking how the government implements comprehensive control measures, highlighting the centralized, state-organized nature of China's response. This state-dominated model extends to specific service domains, where professional personnel undertake critical tasks and their service quality receives attention (e.g., "Training" topic). At the technical level, the community details media coverage on epidemiological responses to HIV/AIDS (e.g., "Infectious diseases", "Disease reporting" topics), including rigorous surveillance and reporting systems, demonstrating media's close attention to epidemic dynamics and transmission routes. Additionally, prevention is highlighted through targeted intervention strategies (e.g., "Free of charge" topic). Free testing and counseling services, along with universal screening for high-risk groups and the general public, constitute focal points in media coverage, aiming to promote early detection of the disease.

**Theme 2.** The second community network centers on HIV/AIDS related publicity campaigns, which we have designated as the "Publicity" theme. Comprising 10 interconnected topics (light green nodes) in Fig 1, this community encompasses diverse forms of knowledge dissemination and social advocacy initiatives. Direct publicity and educational activities (e.g., "Publicity", "HIV/AIDS education" topics) serve as the foundation, with media frequently covering both online and offline events. Such public communication often incorporates philanthropic elements (e.g., "Public welfare" topic), serving as important vehicles for raising social awareness.

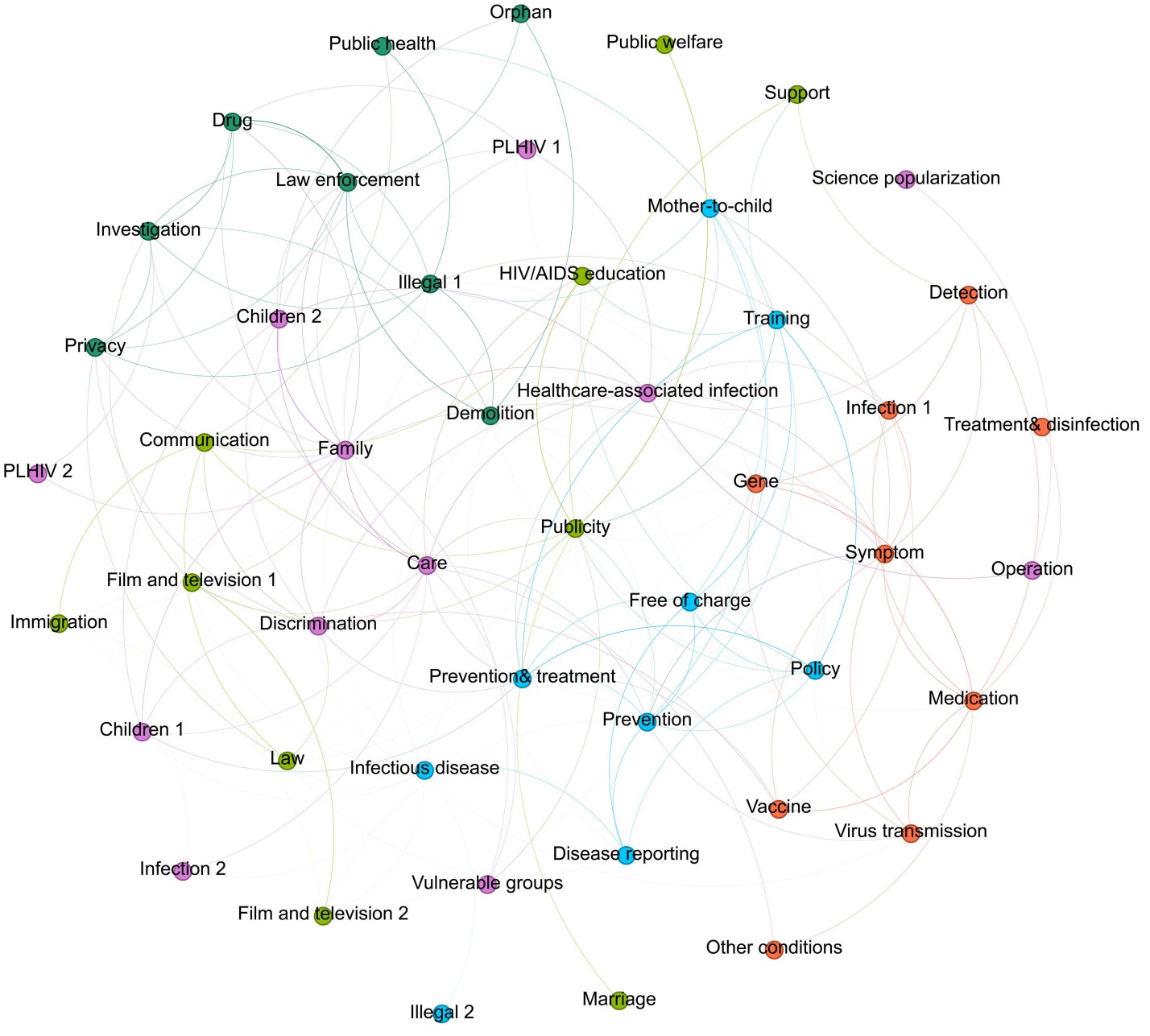

**Fig 1. Topic network of HIV/AIDS news.**

Media coverage within this thematic community also demonstrates the diversity and contextuality of publicity approaches. Utilizing film and television productions (e.g., "Film and television" topic) for advocacy is a prominent feature, aiming to deepen public understanding through vivid narratives. Concurrently, publicity extends to specific groups and settings, such as HIV/AIDS education targeting student populations and marriage-related contexts (e.g., "Marriage" topic). Notably, the media also address social norms and legal ethics (e.g., "Law" topic). Furthermore, interactive exchange

activities (e.g., "Communication" topic) and various competitions organized centered on HIV/AIDS prevention and control knowledge have also enriched the forms of publicity and promoted public participation.

**Theme 3.** The third community network focuses on social issues, which we have designated as the "Society" theme. This theme consists of eight interconnected topics in the network shown in Fig 1 (represented by dark green nodes). The core issues in this category of coverage revolve around the intersection of HIV/AIDS with various social behaviors and public governance, primarily presenting two main threads: First, illegal and criminal behaviors and their law enforcement responses—including drug abuse (e.g., "Drugs" topic) and various illegal activities (e.g., "Illegal", "Demolition", and "Law enforcement" topics). For instance, intimidation using PLHIV status during demolition activities and spreading rumors to disrupt social order have triggered strong intervention from public security and health departments. Second, complex social management dilemmas involving specific populations, include the issue of orphans (e.g., "Orphans" topic), along with privacy infringement and information leakage that have emerged with the internet age (e.g., "Privacy" topic). This structure highlights the challenges that HIV/AIDS issues pose to grassroots social governance and the implementation of laws and regulations.

**Theme 4.** The fourth community network centers on HIV/AIDS-related medical information. Accordingly, we have designated this theme as the "Medicine," comprising nine interconnected topics in the network depicted in Fig 1 (represented by orange nodes). This community initially focuses on foundational aspects of medical cognition of HIV/AIDS (e.g., "Virus transmission" and "Infection" topics). In the intermediate and advanced stages of disease management, the media frequently address therapeutic approaches (e.g., "Medication" and "Symptom" topics), while also encompassing supportive fundamental safeguards (e.g., "Treatment and disinfection" topic). Notably, the media demonstrates sustained attention to cutting-edge medical exploration. On one hand, breakthroughs in gene therapy (e.g., "Gene" topic) and their associated potential for cures receive significant coverage. On the other hand, vaccine development (e.g., "Vaccine" topic) has emerged as a focal point for global collaboration and intensive scientific research. Overall, the "Medicine" themed community covers scientific explorations ranging from infection mechanisms, diversified treatment plans to cutting-edge scientific breakthroughs. This reflects the sustained concern of Chinese society for the development of medical knowledge and the expectation of ultimately overcoming it.

**Theme 5.** The fifth community network focuses on the content related to people living with HIV, which we have designated as the "PLHIV" theme. This theme comprises twelve interconnected topics in the network depicted in Fig 1 (represented by purple nodes). This community illustrates the profound impact of HIV/AIDS on individuals and families (e.g., "Family" topic), with media coverage extensively documenting the difficult decisions faced by PLHIV when disclosing their status to family members and the subsequent effects on their family life trajectories. Furthermore, the media demonstrates heightened attention toward protected or socially marginalized cases (e.g., "Children" and "Vulnerable Groups" topics), with the coverage predominantly focusing on the compounded challenges arising from intersectional identities. Intertwined with survival pressures is the pervasive issue of stigmatization and discrimination (e.g., "Discrimination" topic). Media discourse consistently examines the barriers that social prejudice creates in employment and daily lives, underscoring the necessity of protecting their rights. In the healthcare domain, the coverage within this community highlights the challenges PLHIV face in accessing equitable and available services (e.g., "Operation" and "Healthcare associated infection" topics). Ultimately, the "Care" topic emerges as the central focus of this community, reflecting efforts to construct social support networks.

## Changes over time

Fig 2 illustrates the temporal dynamics of news volume and thematic distribution. Subfigure (a) represents the annual quantity of relevant news articles, while subfigure (b) shows the share of themes within each year's coverage, as quantified by mean topic probabilities derived from ANTMN analysis. In terms of news volume, the coverage peaked in 2011, followed by an overall decline. By 2024, the volume had decreased to less than one-tenth of its peak level.

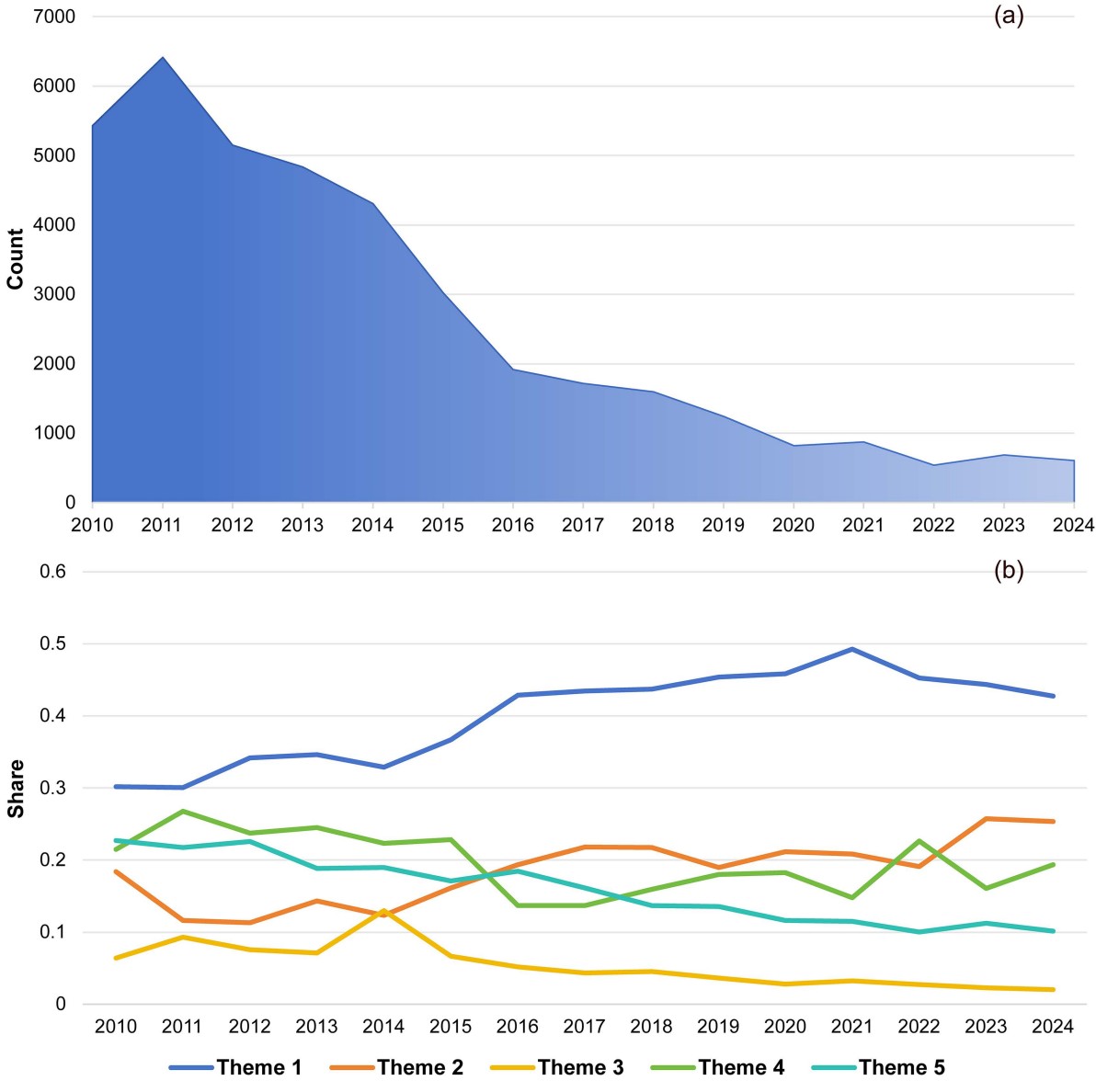

**Fig 2. HIV/AIDS discourse over time.**

To better observe the changes in the share of language across themes, we calculated the temporal trends for each theme using the Mann-Kendall trend test with False Discovery Rate (FDR) correction. Over the past 15 years of news coverage, the "Prevention and control" theme has consistently maintained the highest linguistic share and demonstrated a significant upward trend (Z=3.365, adjusted p=.001). Its slope was steeper than that of the similarly increasing "Publicity" theme (Z=2.969, adjusted p=.003), leading to a markedly widened gap in share compared to other themes after 2014. The linguistic share of the "Publicity" theme rose from its initially low position to second place, indicating its elevated prominence. Themes exhibiting significant downward trends included "Society" (Z=−4.157, adjusted p<.001) and "PLHIV" (Z=−4.652, adjusted p<.001). Only the "Medicine" theme remained stable, showing no significant trend overall (Z=−1.485, adjusted p=.138).

## PLHIV terminology and HIV/AIDS metaphors

We conducted a matching analysis across the entire sample using a predefined glossary of PLHIV terminology and HIV/AIDS metaphors. During this process, given that terminology could represent two distinct categories, the corresponding news items were simultaneously annotated with both category markers. Table 1 presents the proportion of news articles

**Table 1. List and usage rate of PLHIV terminology.**

| PLHIV terminology | Usage Rate | Chinese examples (Translation) |
|---|---|---|
| Generic | 55.95% | aizibing huanzhe (AIDS Patient)<br>aizi bingdu ganranzhe (HIV-infected Person)<br>aizibing huan (AIDS Victim) |
| Children | 2.85% | aizibing gu'er (AIDS Orphan)<br>aizi ertong (AIDS Child)<br>aizi bingdu ying'er (AIDS Infant) |
| Females | 1.00% | aizi mama (AIDS Mother)<br>aizi xinniang (AIDS Bride)<br>aizi yunfu (AIDS Pregnant Woman) |
| Criminals | 0.78% | aizibing zuifan (AIDS Criminal)<br>aizi xiaotou (AIDS Thief)<br>aizi jiefei (AIDS Robber) |
| Youths | 0.77% | aizi shaonian (AIDS Teenager)<br>aizi nvhai (AIDS Girl)<br>aizibing nantong (AIDS Boy) |
| Elderly | 0.58% | aizi gulao (AIDS Orphaned Elder)<br>aizi laoren (AIDS Senior)<br>gaoling aizi (AIDS Elder) |
| Social roles | 0.53% | aizi laoshi (AIDS Teacher)<br>aizi kaosheng (AIDS Candidate)<br>aizibing qunzhong yanyuan (AIDS Actor) |
| Spatialization | 0.27% | aizi gongyu (AIDS Apartment)<br>aizi xiaozhen (AIDS Town)<br>aizi cun (AIDS Village) |
| Males | 0.27% | aizi xiaohuo (AIDS Guy)<br>aizi laofuqin (AIDS Father)<br>ran aizi nanxing (AIDS Male) |
| Homosexuals | 0.18% | nantong aizi (AIDS Homosexual) |
| Kin | 0.15% | aizi jiating (AIDS Family)<br>aizi fuzi (AIDS Father and Son)<br>aizibing danqin jiating (AIDS Single-Parent Family) |
| Partners | 0.08% | aizi bingdu tongbao (AIDS Compatriot)<br>aizi bingyou (AIDS Friend) |
| Couples | 0.06% | aizi fufu (AIDS Couple)<br>aizi fuqi (AIDS Husband and Wife) |
| Bastards | < 0.05% | aizi lao (AIDS Bastard) |
| Underprivileged | < 0.05% | aizibing kunnanhu (AIDS Poor People)<br>aizibing tekunhu (AIDS Extremely Poor People)<br>canjiren aizi (AIDS Disabled Person) |
| Overseas Chinese | < 0.05% | aizi huaren (AIDS Overseas Chinese) |
| Sex workers | < 0.05% | aizi maiyinnv (AIDS Prostitute) |
| Role model | < 0.05% | aizibing dianfan (AIDS Role Model) |
| Foreigners | < 0.05% | liuxue renyuan aizi bingdu (AIDS International Student) |

Note: Values less than 0.05% are denoted as < 0.05%.

employing PLHIV terminology, with three collocation examples provided for each terminology category (certain categories contain fewer than three collocations).

In the PLHIV terminology, "Generic" terminology demonstrated clear predominance, employed by 55.95% of the analyzed news articles. By contrast, stigmatizing terminology targeting specific population groups exhibited significantly lower usage. Within the stigmatizing terminology, "Children" and "Females" showed relatively higher intra-category prevalence.

According to the proportion data of news articles employing HIV/AIDS-related metaphorical collocations (see Table 2), war metaphors constituted the most extensively employed type, ranking highest in prevalence among all metaphor categories. Together with journey metaphors and entity metaphors, they formed the core metaphorical framework.

MCA was conducted to examine the primary trends of co-occurrence between discursive strategies and news themes. For PLHIV terminology (Fig 3), the first two dimensions yielded eigenvalues of 0.0709 and 0.0271, explaining 65.55% and 25.01% of the total variance, respectively (cumulative explained variance: 90.56%). For HIV/AIDS metaphors (Fig 4), the first two dimensions yielded eigenvalues of 0.2456 and 0.0821, explaining 71.79% and 23.99% of the total variance, respectively (cumulative explained variance: 95.78%). Given that this study focuses on the primary trends of co-occurrence between PLHIV terminology, metaphors, and themes, and considering that all variables are binary, a two-dimensional solution is sufficient to represent the primary structure of the data. However, it should be emphasized that the MCA results for both reflect only the major trends rather than detailed information. Specifically, regarding terminological choices, the "Society" theme demonstrated a predominant association with "Females" terminology, while the "PLHIV" theme was strongly correlated with terminology such as "Children", "Kin", and "Couples". In contrast, terminological choices for the "Prevention and control", "Publicity", and "Medicine" themes exhibited considerable convergence, primarily linked to terminology like "Generic", "Youths", and "Elderly".

Concerning metaphorical choices, the "Prevention and control" and "Publicity" themes showed stronger associations with war, movement, journey, and architectural metaphors. The "Medicine" theme, conversely, was primarily associated with pollution, humans, and war metaphors. The "PLHIV" and "Society" themes, comparatively, demonstrated connections with entity, nature, monster, and container metaphors.

## Co-occurrence patterns

Furthermore, we investigated the co-occurrence patterns between terminological and metaphorical strategies. Our findings indicate that at the standard significance threshold, no significant network was formed among the majority of terminology and metaphors, suggesting that the choice of terminological and metaphorical strategies is largely independent. However, we also identified a small number of cross-strategy components that exhibited significant internal associations. The internal structures and connection weights of these components are presented in Table 3. Specifically, "Generic" terminology showed significant connections to both the war and journey metaphors, while "Children" terminology demonstrated a significant co-occurrence with the entity metaphors.

**Table 2. List and usage rate of HIV/AIDS metaphors.**

| HIV/AIDS Metaphors | Usage Rate | HIV/AIDS Metaphors | Usage Rate |
|---|---|---|---|
| War metaphors | 9.06% | Pollution metaphors | 0.17% |
| Journey metaphors | 6.93% | Human metaphors | 0.08% |
| Entity metaphors | 3.75% | Architectural metaphors | < 0.05% |
| Movement metaphors | 1.68% | Container metaphors | < 0.05% |
| Nature metaphors | 0.31% | Apocalyptic metaphors | < 0.05% |
| Monster metaphors | 0.21% | Animal metaphors | < 0.05% |

Note: Values less than 0.05% are denoted as < 0.05%.

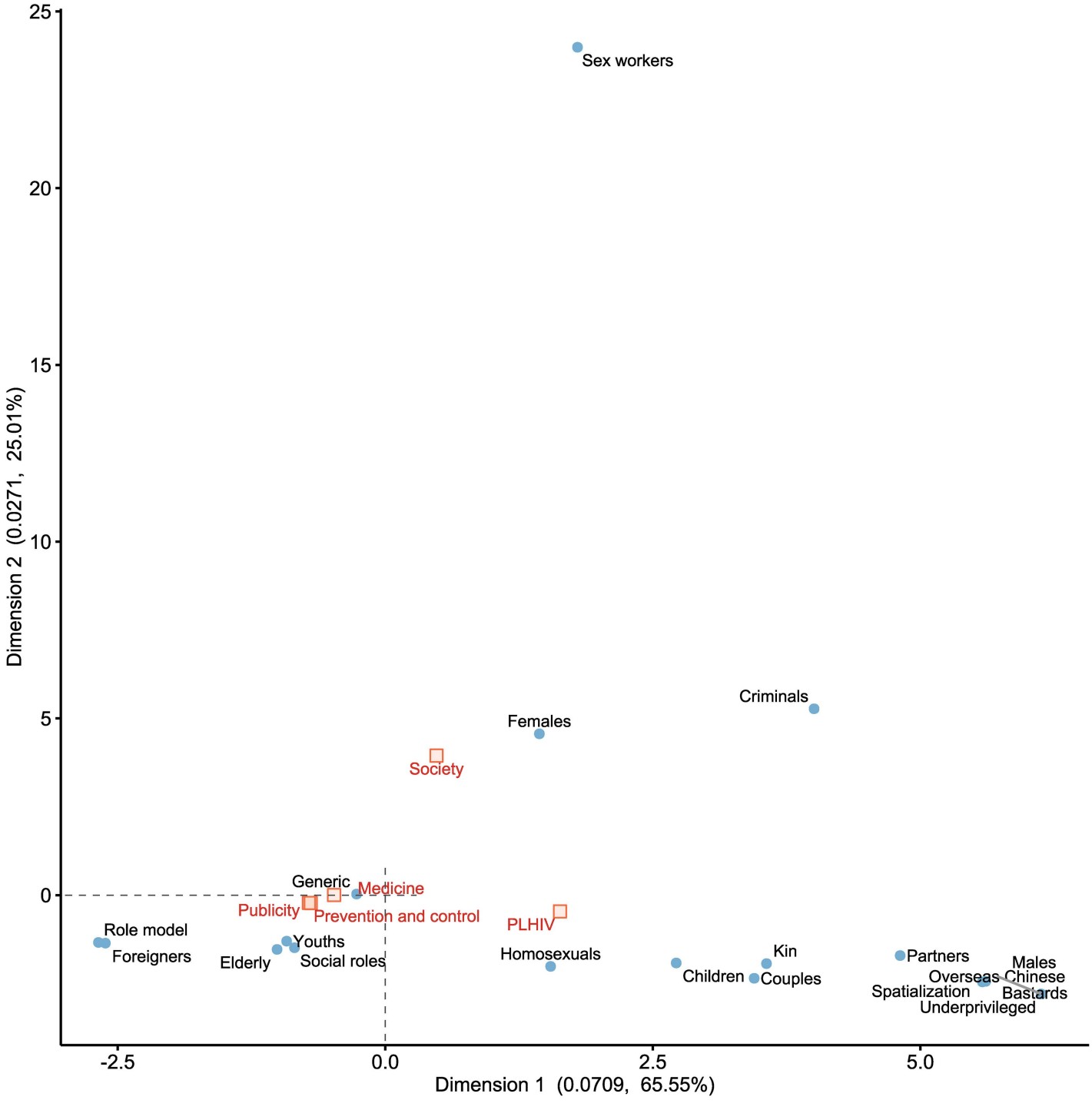

**Fig 3. PLHIV terminology and news themes.** The Fig 3 is based on the first two dimensions derived from the correspondence analysis. The numerical value on the axis represents the eigenvalue, and the percentage indicates the percentage of contribution rate of that dimension. The positions of the points reveal their associations within the two-dimensional semantic space.

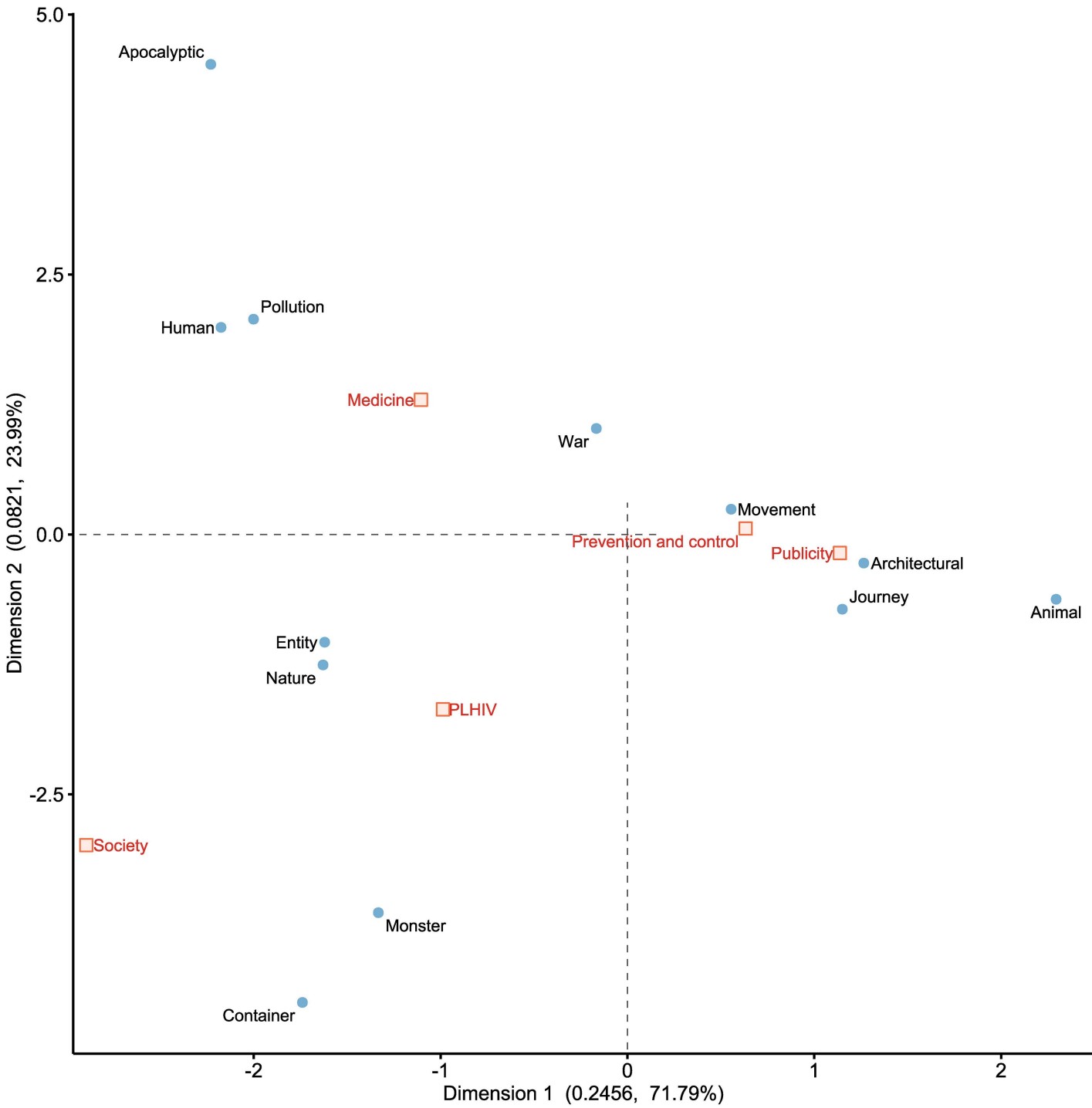

**Fig 4. HIV/AIDS metaphors and news themes.** The Fig 4 is based on the first two dimensions derived from the correspondence analysis of metaphors. The numerical value on the axis represents the eigenvalue, and the percentage indicates the percentage of contribution rate of that dimension. The positions of the points reveal their associations within the two-dimensional semantic space.

**Table 3. Co-occurrence across discursive strategy types.**

| Co-occurrence | Weight | P value |
|---|---|---|
| Generic — War metaphors | 0.10 | <.001 |
| Generic — Journey metaphors | 0.09 | <.001 |
| Children — Entity metaphors | 0.06 | .007 |

To further unpack the relationships between metaphors and terminology designating specific populations, we excluded non-specific "Generic" terminology and directed our subsequent analysis exclusively toward the significant pairing between "Children" terminology and the entity metaphors. We found that a total of 148 news reports simultaneously employed "Children" terminology and the entity metaphors. Table 4 summarizes the source domain elements of the entity metaphors identified within these reports. The term *xiedai* (carry HIV/AIDS) overwhelmingly dominated the discourse, characterized by "Agentive", "Comitative", and "Mobile" source elements. In contrast, alternative expressions such as *qieduan* (cut off) and *daiyou* (have) appeared marginally, indicating a strong linguistic preference for agentive and mobile representations over patientive ones.

## Discussion

### Major findings

This study identifies five primary themes: "Prevention and control", "Publicity", "Society", "Medicine", and "PLHIV", encompassing multidimensional perspectives on HIV/AIDS ranging from macro-level policies to individual experiences. Specifically, the "Prevention and control" theme captures the state-led, policy-driven system of HIV governance, including national strategies, surveillance, and targeted interventions. The "Publicity" theme highlights the crucial role of information dissemination and social awareness campaigns. The "Society" theme highlights how HIV/AIDS intersects with social order and governance, focusing on crime, law enforcement, and vulnerable groups. The "Medicine" theme delves into the scientific understanding and therapeutic advancements of the disease, and the "PLHIV" theme centers on the lived realities, struggles, and needs of individuals affected by HIV/AIDS. By dissecting these thematic networks, this research offers a comprehensive overview of media discourse on HIV/AIDS, providing insights into the dominant narratives, evolving concerns, and public perception and societal response to the epidemic.

Our analysis reveals shifts in the thematic landscape over the past 15 years. While overall news volume peaked in 2011 and subsequently declined, the distribution of themes reveals more nuanced trends. Notably, the "Prevention and control" theme has dominated consistently over the past 15 years, with its prominence increasing significantly over time. This trend underscores China's state-dominated model in HIV/AIDS response, aligning with existing research that highlights the country's reliance on top-down communication, national resource allocation and policy-driven initiatives [2,12].

**Table 4. Metaphor elements of entity metaphors in news using "Children" terminology.**

| Chinese Metaphors and Translation | Elements | News Count |
|---|---|---|
| xiedai (carry) | Agentive; Comitative; Mobile | 144 |
| qieduan (cut off) | Agentive; Dynamic; Connective | 3 |
| daiyou (have) | Patient; Static; Comitative | 1 |

Note: This table presents the source domain elements of different ontological metaphorical expressions found in news reports containing both "Children" terminology and ontological metaphors (148 articles in total). The frequency in the rightmost column represents the number of news articles that employ each corresponding ontological metaphor expression.

Concurrently, the "Publicity" theme, though initially less prominent, has risen to second place, indicating the growing importance of information dissemination, health education and policy advocacy in the media agenda. This reflects a common practice in developing countries, where mass media are leveraged to reduce the negative impacts of HIV/AIDS [8].

By contrast, while the share of the "Prevention and control" theme has grown steadily, the prominence of the "Society" and "PLHIV" themes in media discourse has markedly declined over the years. This shift reveals a narrative transition from emphasizing social conflicts, individual predicaments, and PLHIV stories toward highlighting the efficient operation of state mechanisms and policy implementation. Such policy-driven discourse not only mirrors the practical realities of HIV/AIDS governance in China but also exemplifies a pro-government framing, standing in stark contrast to the anti-government framing prevalent in Western media [13]. This dichotomy highlights how media roles in public health discourse are shaped by distinct sociopolitical systems. By contrast, the "Medicine" theme remained stable throughout the 15-year period. Such stability is rooted in its consistent coverage of both medical knowledge and disease management strategies, reflecting the enduring core concerns of medical practice over time [31]. Despite this enduring thematic focus, China's media framing of HIV/AIDS is not static but dynamically evolves in response to the shifting interplay between national policy priorities and epidemiological developments.

A major empirical contribution of this study is the granular identification of understudied stigmatizing terminology, made possible by our large-scale, 15-year dataset. The choices of terminology regarding PLHIV reflect cognitive characteristics marked by the coexistence of de-identification and stigmatization. On one hand, the overwhelming predominance of "Generic" terminology reveals a de-identified discursive strategy. On the other hand, de-identification does not equate to stigma eradication. This study reveals that stereotypical labels persist and have even diversified beyond prior findings. For instance, we identified the persistent use of criminalizing and marginalizing labels such as "Criminals" and "Underprivileged", as well as social role-based stigmatizing designations tied to modern societal structures, such as "AIDS Teacher" and "AIDS Candidate". More importantly, we presented unique discoveries regarding the spatialization of stigma. While earlier studies noted the term "AIDS Village" [2], our data reveals a critical evolution and diversification of spatial stigma through terms like "AIDS Apartment" and "AIDS Town". This transition from rural "Village" to urban "Apartment" and "Town" reflects the urbanization of the HIV/AIDS narrative, demonstrating that spatial stigma has not diminished; rather, it has dynamically adapted to contemporary geography. Such terminology directly associates specific geographical areas with HIV/AIDS, reducing infected populations to mere geographical units. Consequently, they construct visibly demarcated "quarantine zones" in public perception, further exacerbating social exclusion and discrimination.

Furthermore, this study conducted a multidimensional analysis of the associations among thematic contexts, PLHIV terminology, and HIV/AIDS metaphors. While previous literature has frequently analyzed these elements in isolation, the MCA elucidates how macro-level thematic contexts dictate micro-level linguistic choices. This intersectional approach is instrumental in uncovering deeper narrative biases. For example, the "Society" theme exhibits a high degree of association with "Females" terminology, suggesting a gendered narrative tendency in HIV/AIDS social discourse and the potential for gender discrimination. In contrast, the "PLHIV" theme prominently features identity-based terminology such as "Children", "Kin", and "Couples", highlighting its emphasis on relational and demographic characteristics. Rather than positioning the PLHIV as an isolated clinical case, this narrative situates PLHIV within extended networks of care, obligation, and intimacy. It shifts attention from clinical markers to the distribution of impact, responsibility, and care across households and intimate partnerships. Notably, the three macro-level themes—"Prevention and control", "Publicity", and "Medicine"—demonstrate a convergence in favoring "Generic", "Youths", and "Elderly" terminology, reflecting a depersonalized discursive strategy in institutional health communication.

Similarly, analyzing the differing use of metaphors across various themes can provide richer perspectives. Our novel contribution is demonstrating how these metaphors are compartmentalized by themes to serve distinct governance functions. The "Prevention and control" and "Publicity" themes collectively employ war, movement, and architectural metaphors to construct a defense-oriented cognitive model. The "Medicine" theme predominantly relies on pollution and human metaphors to reinforce a pathogen-centric perspective. Most notably, we identified a distinctive metaphorical

configuration within the "PLHIV" and "Society" themes, comprising entity, nature, monster, and container metaphors. This configuration serves a dual function: while entity and nature metaphors operate as relatively neutralizing devices conducive to destigmatization, monster and container metaphors risk reinforcing social boundaries and "othering" narratives.

The tension between destigmatizing and "othering" narratives highlights the gap between entrenched media language and evolving clinical realities. Historically, the predominance of war metaphors aligned with early epidemic anxieties [27]. However, with medical advancements, HIV/AIDS has transformed into a manageable chronic condition. In our study, we found that metaphors with positive narrative frames are replacing war metaphors, with journey metaphors being the most prevalent. Movement and nature metaphors also exceeded monster and pollution metaphors that frequently appeared in earlier narratives, as identified by Sontag [27]. Journey metaphors conceptualize living with HIV/AIDS as a journey, emphasizing agency in treatment and recovery. Entity metaphors concretize the disease into tangible, measurable, and manageable entities, focusing more on control and coexistence rather than the adversarial stance of war metaphors. Highly confrontational terms such as "attack" and "defense" not only disconnect from the lived realities of PLHIV but also exacerbate stigma, exclusion, and discrimination [29]. Consequently, alternative metaphors with positive narrative frameworks have emerged as more constructive options [59].

Finally, our co-occurrence analysis has revealed significant linkages between specific terminology and metaphors. While "Generic" terminology frequently co-occurred with war and journey metaphors to facilitate de-identification, stigmatizing "Children" terminology strongly aligned with entity metaphors. This stigmatization is predominantly driven by the term *xiedai* (carry), which emphasizes "Agentive" and "Comitative" traits (e.g., "the news of a baby girl born carrying HIV once caused a sensation"). This linguistic choice objectifies children as mere vessels, conceptualizing HIV as a movable entity and children as active agents. Consequently, this framing unintentionally amplifies societal stigma by overshadowing their reality as vulnerable patients. It engenders a cognitive bias that shifts the focus from their need for care to their perceived threat as mobile vectors for viral transmission.

In summary, drawing on a 15-year longitudinal dataset, this study reveals a complex coexistence of de-identification and stigma diversification. With "Generic terms" accounting for 55.95% of the usage, our findings illuminate active destigmatization efforts that have been largely obscured by prior literature's heavy focus on stigmatization [3]. Crucially, moving beyond the limitations of traditional manual sampling and descriptive statistics, our computational approach enabled the detection of highly latent, low-frequency discursive patterns that conventional methods would inevitably overlook. This methodological advantage produces substantively new insights. By identifying undocumented stigmatizing terms related to social roles and spatialization, such as "AIDS Teacher" and "AIDS Apartment", we elevate our analysis to a theoretical advancement. This fills previous research gaps by showing that stigma now permeates modern social structures rather than just targeting traditional marginalized groups [2]. Furthermore, our multidimensional analysis exposes deeper narrative biases, such as gendered associations in the "Society" theme and relational framing in the "PLHIV" theme, which moves beyond prior research that has typically been confined to univariate or bivariate analyses [11]. Finally, we capture a critical discursive shift: while confrontational metaphors persist, positive alternatives—namely journey and entity metaphors—have actively emerged to reflect the modern clinical reality of HIV/AIDS as a manageable chronic condition. This finding challenges the prevailing narrative in previous Chinese HIV/AIDS discourse studies, which largely framed the disease through negative metaphors like war, pollution, and mutation [60,61]. Collectively, these novel findings highlight the dynamic tension between destigmatization efforts and entrenched media biases, providing crucial insights for fostering more inclusive public health communication.

## Theoretical contributions and practical implications

This study systematically maps the narrative frameworks and strategic choices in Chinese media coverage of HIV/AIDS over the past 15 years, offering novel methodological and theoretical perspectives on how the media construct health risk cognition. First, within the field of health communication, our findings illuminate the dynamics of a state-dominated,

policy-driven HIV/AIDS governance system. The analysis underscores how institutional media rely on top-down communication, national resource allocation, and policy-oriented initiatives to shape public health literacy. Second, in the realm of risk communication, this study captures a critical paradigm shift in disease risk framing. By documenting the transition from an adversarial frame characterized by war metaphors to a constructive narrative frame driven by journey metaphors, we provide a novel framework for understanding how risk boundaries are linguistically redefined in modernizing societies. Finally, for media discourse analysis, our large-scale computational approach, paired with intersectional analysis, pushes the field beyond traditional descriptive framing. This methodological advantage enables the detection of highly latent, low-frequency stigmatizing terminology at the margins of the discourse, uncovering nuanced patterns that conventional manual approaches may inevitably overlook.

At the practical level, this study offers valuable implications for public health communication, policy making, and social intervention. Media practitioners should further avoid labels and metaphorical expressions that exacerbate stigmatization. Efforts to mitigate stigma must extend beyond common stereotypical labels to encompass broader contexts. To address the deficiencies identified in traditional media coverage, media professionals need to transform their reporting strategies. First, they should rebalance thematic focus by countering the current overemphasis on macro-level policy narratives and creating regular spaces for the voices and lived experiences of PLHIV, especially children, women, and other vulnerable groups. Second, in response to the persistent and diverse use of stigmatizing labels, media organizations can develop and disseminate guidelines for non-stigmatizing language, offering people-centered alternatives. Third, given the dominance and potential harms of war metaphors, the media should reduce confrontational war framings and adopt more constructive metaphors. These should be combined with narratives emphasizing resilience and recovery, focusing on the agency of PLHIV rather than solely on the threat posed by the virus. Finally, as stigmatizing language is often theme-specific, interventions must be tailored. For example, when reporting on social issues, editors should pay particular attention to avoiding gender stereotypes.

Policymakers need to prioritize the impact of specific terminology on referenced groups, developing more inclusive and targeted protective measures. In terms of social intervention, this study urges Chinese media professionals to shift their focus from macro-level prevention to individual predicaments, and to more proactively fulfill the media's auxiliary role in building social support systems.

Beyond the Chinese context, this study provides broader theoretical insights for global health communication, demonstrating transferability to other settings. Our findings on the shift from confrontational to constructive metaphors suggest that communication strategies are both stage-specific and context-dependent. As epidemics evolve from the outbreak to normalized management, media linguistic strategies correspondingly transition from crisis-oriented to collaboration-oriented approaches. This dynamic evolution provides a predictive framework for understanding media discourse of other global infectious diseases transitioning toward normalization. Furthermore, with continued global development and urbanization, HIV/AIDS stigmatization terminology is being reconstructed. Stigmatization has expanded from traditional marginalized groups to modern social structures, evidenced by new forms like "AIDS Teacher" and "AIDS Candidate", and spatial shifts from rural to urban contexts like "AIDS Apartment" and "AIDS Town". This provides a transferable analytical framework for exploring how evolving socioeconomic conditions reshape stigma language across different cultural contexts.

## Limitations and future research

Despite its contributions, this study has several limitations. First, the sampling frame is restricted to a specific subset of traditional media. This focus may introduce sampling frame bias, as it does not fully capture the diversity of contemporary Chinese media practices. For instance, authoritative state-aligned media and market-oriented commercial outlets may employ divergent narrative strategies and terminology. Future research should therefore expand the sampling frame to include a wider range of media types, and employ probabilistic or stratified sampling to enhance representativeness.

Second, focusing exclusively on traditional media neglects a systematic analysis of media heterogeneity. Social media platforms like Weibo and Douyin host fragmented, personalized HIV/AIDS discourses that differ markedly from institutionalized narratives. Future research should fill this gap via comparative analyses to examine how diverse digital communities appropriate and reframe HIV/AIDS terminology. Additionally, longitudinal cross-media studies could track whether stigmatizing frames converge or diverge between traditional and social media over time.

Finally, although the collocation-based method efficiently processed the 15-year dataset, the coding still involves subjective judgments in constructing the glossary of PLHIV terminology and HIV/AIDS metaphors, which may miss localized or emerging expressions. The rule-based approach also has limited capacity to detect highly implicit or complex metaphors beyond typical collocation patterns. Future studies could adopt deep learning and large language models for context-aware semantic analysis, enabling automatic detection of nuanced stigmatization beyond predefined lexicons.

## Conclusion

This study systematically examines the thematic structures, discursive strategies, and cognitive implications of HIV/AIDS coverage in Chinese traditional media over the past 15 years. The findings reveal a media landscape centered on the "Prevention and control" theme, with multidimensional coexistence of the "Publicity", "Society", "Medicine", and "PLHIV" themes. Over this period, the prominence of the "Prevention and control" theme has steadily intensified, while that of the "Society" and "PLHIV" themes have diminished, widening the disparity between macro-level narratives and individual-focused narratives. Regarding terminology for PLHIV, de-identified "Generic" terminology has become mainstream, yet stigmatizing terminology persists and exhibits typological diversity. Metaphorically, while war metaphors dominate, alternatives such as journey metaphors and entity metaphors hold significant presence, reflecting explorations toward more constructive discourse. The differential configurations of PLHIV terminology and HIV/AIDS metaphors across thematic contexts simultaneously reflect strategic adaptations in media communication tailored to specific issues and intended audiences, while also revealing the entrenched inertia of stigmatization. We recommend that the media further refine labeling language and inflammatory metaphors. Policies should establish more comprehensive rhetorical guidelines and standardize language usage, while advocating for an appropriate shift in media coverage focus from macro-level narratives to individual narratives.

## Supporting information

**S1 File. News matrix and topic words.**
(ZIP)

## Author contributions

**Conceptualization:** Haiyong Shi.

**Data curation:** Yuhang Li, Lisai Yu, Haiyong Shi.

**Formal analysis:** Chenghui Wu, Lisai Yu.

**Investigation:** Chenghui Wu, Lisai Yu.

**Methodology:** Yuhang Li.

**Supervision:** Haiyong Shi.

**Validation:** Yuhang Li.

**Visualization:** Lisai Yu.

**Writing – original draft:** Yuhang Li.

**Writing – review & editing:** Haiyong Shi.

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
