## [Decision Letter · Decision Letter 0]

23 Dec 2025

PONE-D-25-52841Constructing HIV/AIDS in Chinese Media (2011-2024): A Mixed-Methods StudyPLOS One

Dear Dr. Shi,

Thank you for submitting your manuscript to PLOS ONE. After careful consideration, we feel that it has merit but does not fully meet PLOS ONE’s publication criteria as it currently stands. Therefore, we invite you to submit a revised version of the manuscript that addresses the points raised during the review process.

We look forward to receiving your revised manuscript.

Kind regards,

Xiaoming Tian, Ph.D.

Academic Editor

PLOS One

Journal Requirements:

3. Please upload a new copy of Figure 1 as the detail is not clear. Please follow the link for more information:  https://journals.plos.org/plosone/s/figures

4. Thank you for providing your underlying data as Supporting Information.

We note that the data set contains text or data that is not in English. Please note that PLOS is an English-language publisher, so we require data sets to be provided in English as well. Please upload an English-language version of your data set.

This will also allow us to determine if your data follows PLOS standards per our Data Availability policy here: https://journals.plos.org/plosone/s/data-availability

5. We note that there is identifying data in the Supporting Information file <AIDS Corpus.xlsx>. Due to the inclusion of these potentially identifying data, we have removed this file from your file inventory. Prior to sharing human research participant data, authors should consult with an ethics committee to ensure data are shared in accordance with participant consent and all applicable local laws.

-Location data

Please remove or anonymize all personal information (Names), ensure that the data shared are in accordance with participant consent, and re-upload a fully anonymized data set. Please note that spreadsheet columns with personal information must be removed and not hidden as all hidden columns will appear in the published file.

Reviewers' comments:

Reviewer's Responses to Questions

**Comments to the Author**

1. Is the manuscript technically sound, and do the data support the conclusions?

Reviewer #1: Yes

Reviewer #2: Partly

2. Has the statistical analysis been performed appropriately and rigorously? 

Reviewer #1: Yes

Reviewer #2: Yes

3. Have the authors made all data underlying the findings in their manuscript fully available?

Reviewer #1: Yes

Reviewer #2: Yes

4. Is the manuscript presented in an intelligible fashion and written in standard English?

Reviewer #1: Yes

Reviewer #2: Yes

5. Review Comments to the Author

Reviewer #1: This study examines 39,141 news articles published between 2010 and 2024 to understand the evolving media discourse surrounding HIV/AIDS. Employing Analysis of Topic Model Networks (ANTMN) and collocation analysis, we extract the thematic networks, terminology for people living with HIV (PLHIV), and HIV/AIDS metaphors. Multiple Correspondence Analysis (MCA) and network backbone method were complementarily employed to examine the interrelationships among discursive strategies and their connections with thematic contexts. In my opinion, this is an excellent paper characterized by rigorous logic, sound methodology, proficient English expression, reliable data, and credible conclusions. However, there are several minor issues that.

Imprecise language expression:

The phrase "particularly in contexts like China" is mentioned in lines 49–50. Please briefly elaborate on why China is specifically highlighted in this context.

Formatting inconsistencies:

In line 74, two references are enclosed in square brackets with a comma between them, which does not comply with the formatting requirements; similar issues exist elsewhere in the text.

An extra comma is present in line 133.

The meaning of decimals and percentages next to the axes in Figures 3 and 4 is unclear—annotations should be provided to clarify these.

Unclear research significance:

It is recommended that the authors carefully elaborate, before the methodology section, on the significance of studying traditional media in the current era of highly prevalent social media. Specifically, please clarify: "What is the importance of researching traditional media when social media is so dominant?"

Superficial discussion on addressing deficiencies:

While the paper identifies certain deficiencies in traditional media’s coverage of HIV-related topics, the discussion section only offers general remarks on how to improve these shortcomings. The authors are advised to provide targeted countermeasures and suggestions tailored to these specific deficiencies.

Reviewer #2: Firstly, this manuscript is technically sound and the data could support most part of the conclusion,and the statistical analysis has been performed relatively appropriately. However, for some part of data presented in the Result section, I couldn't find the further analysis and explanation in Discussion section. For example, when discussing the themes identified in this research, the fourth one "Medicine“ is not mentioned in "Key finding" part. For another example, the second category of the result presented is "Changes over time", but I didn't find any further analysis and explanation on this in discussion section. I don't know whether it is one of the major findings of this research. Secondly, as far as I could see, all data collected in this research (including figures and tables) has been included at the end of the manuscript. Thirdly, as an academic paper, this manuscript is presented with the logical structure and in an intelligible fashion. The language is generally clear, correct and unambiguous, but as for the subtitle "Key Findings", "Major Findings" may be more accepted substitute.

6. PLOS authors have the option to publish the peer review history of their article (what does this mean?). If published, this will include your full peer review and any attached files.

Reviewer #1: **Yes:** He Hu

Reviewer #2: No

---

## [Author Response · Author response to Decision Letter 1]

4 Jan 2026

Reviewer #1:

This study examines 39,141 news articles published between 2010 and 2024 to understand the evolving media discourse surrounding HIV/AIDS. Employing Analysis of Topic Model Networks (ANTMN) and collocation analysis, we extract the thematic networks, terminology for people living with HIV (PLHIV), and HIV/AIDS metaphors. Multiple Correspondence Analysis (MCA) and network backbone method were complementarily employed to examine the interrelationships among discursive strategies and their connections with thematic contexts. In my opinion, this is an excellent paper characterized by rigorous logic, sound methodology, proficient English expression, reliable data, and credible conclusions. However, there are several minor issues that.

Response: We are grateful for the reviewer's valuable feedback. In the revised manuscript, we have highlighted the changes in yellow.

1.Imprecise language expression:

The phrase "particularly in contexts like China" is mentioned in lines 49–50. Please briefly elaborate on why China is specifically highlighted in this context.

Response: We appreciate reviewer’s inquiry about the specific mention of China. We have revised the text to clarify why China is specifically highlighted. We now note that HIV/AIDS has become the main cause of death in infectious diseases in China, and the number of high-risk groups is growing. In this case, it is particularly challenging to achieve the public health goal of eliminating HIV/AIDS. See lines 48-53 in Introduction.

2.Formatting inconsistencies:

In line 74, two references are enclosed in square brackets with a comma between them, which does not comply with the formatting requirements; similar issues exist elsewhere in the text.

An extra comma is present in line 133.

The meaning of decimals and percentages next to the axes in Figures 3 and 4 is unclear—annotations should be provided to clarify these.

Response:

Thank you for pointing out these issues. We have carefully checked and revised the manuscript accordingly. The reference formatting (including the case in line 74 and similar instances elsewhere) has been corrected to comply with the journal’s style. The extra comma in line 133 has been removed. In addition, we have added annotations to Figures 3 and 4, respectively, to clarify the meaning of the axis labels. The numerical values represent the eigenvalues, and the percentages indicate the contribution rate of the corresponding dimension. The position of each point reveals its association with others within the two-dimensional semantic space. The complete annotations can be found in the captions of Figures 3 and 4. See lines 410-414 and 421-425 in Section 3.3

3.Unclear research significance:

It is recommended that the authors carefully elaborate, before the methodology section, on the significance of studying traditional media in the current era of highly prevalent social media. Specifically, please clarify: "What is the importance of researching traditional media when social media is so dominant?"

Response:

We appreciate reviewer’s insightful comment regarding the significance of studying traditional media in the age of social media. In the revised manuscript, we emphasize that traditional media still plays a central role in setting agendas and framing public perspective. We also highlight that traditional media discourse often provides reference points that other platforms react to or build upon. These additions, placed before the methodology section, clarify the research significance and justify our focus on traditional media. See lines 107-113 in Introduction.

4.Superficial discussion on addressing deficiencies:

While the paper identifies certain deficiencies in traditional media’s coverage of HIV-related topics, the discussion section only offers general remarks on how to improve these shortcomings. The authors are advised to provide targeted countermeasures and suggestions tailored to these specific deficiencies.

Response:

We appreciate the reviewer’s valuable comment. We agree that the original discussion of how to address deficiencies in traditional media coverage was too general and did not sufficiently translate our empirical findings into concrete, targeted recommendations. In the revised manuscript, we therefore added a paragraph that directly links the main deficiencies identified in our analysis to corresponding actionable recommendations. Specifically, we address the overemphasis on macro-level, policy-oriented narratives by recommending a rebalancing of thematic focus, creating regular spaces for the voices and lived experiences of PLHIV. We respond to the persistent and diversified use of stigmatizing labels by proposing that media organizations develop and disseminate a terminology guide with people-centered alternatives. We tackle the dominance and potential harms of war metaphors by encouraging a shift toward more constructive metaphors, combined with narratives emphasizing resilience, recovery. Finally, in light of the theme specific use of stigmatizing language, we emphasize the need for tailored interventions. These targeted suggestions have been incorporated into the Discussion section. See lines 586-600 in Section 4.2.

Reviewer #2: Firstly, this manuscript is technically sound and the data could support most part of the conclusion,and the statistical analysis has been performed relatively appropriately.

Response:

We are grateful for the reviewer's valuable feedback. In the revised manuscript, we have highlighted the changes in green.

1.However, for some part of data presented in the Result section, I couldn't find the further analysis and explanation in Discussion section. For example, when discussing the themes identified in this research, the fourth one "Medicine“ is not mentioned in "Key finding" part.

Response:

We are grateful for your positive feedback on our manuscript’s technical aspects and statistical analysis. We acknowledge that the "Medicine" theme, despite being identified as a primary theme, was not explicitly discussed in the "Key Findings" section. To address this, we have now incorporated a detailed discussion of the "Medicine" theme, including its thematic composition and temporal stability into the revised "4.1 Major Findings" section. See Section 4.1.

2.For another example, the second category of the result presented is "Changes over time", but I didn't find any further analysis and explanation on this in discussion section. I don't know whether it is one of the major findings of this research.

Response:

Thank you for your insightful comment. In the original manuscript, we did not sufficiently highlight the discussion of changes over time. In the revised version, we now discuss in more detail the trends and possible reasons for the changes in all five themes over time, including the rising prominence of “Prevention and control” and “Publicity,” the decline of “Society” and “PLHIV,” and the relative stability of “Medicine.” See Section 4.1.

3.Secondly, as far as I could see, all data collected in this research (including figures and tables) has been included at the end of the manuscript. Thirdly, as an academic paper, this manuscript is presented with the logical structure and in an intelligible fashion. The language is generally clear, correct and unambiguous, but as for the subtitle "Key Findings", "Major Findings" may be more accepted substitute.

Response: Thank you for your valuable suggestion. We have revised "Key Findings" to "Major Findings".

---

## [Decision Letter · Decision Letter 1]

2 Mar 2026

PONE-D-25-52841R1Constructing HIV/AIDS in Chinese Media (2011-2024): A Mixed-Methods StudyPLOS One

Dear Dr. Shi,

Thank you for submitting your manuscript to PLOS ONE. After careful consideration, we feel that it has merit but does not fully meet PLOS ONE’s publication criteria as it currently stands. Therefore, we invite you to submit a revised version of the manuscript that addresses the points raised during the review process.

We look forward to receiving your revised manuscript.

Kind regards,

Xiaoming Tian, Ph.D.

Academic Editor

PLOS One

Journal Requirements:

Additional Editor Comments:

I now have time to read the revised manuscript carefully and the comments from the three reviewers. After a thorough review of the revised manuscript and the feedback from all three reviewers, I have determined that this manuscript requires major revision before it can be considered for publication.

The core concerns raised by the reviewers, particularly the reviewer who recommended rejection, highlight critical areas that need to be addressed to meet the journal’s publication standards.

First, you need to clearly articulate the novel contribution of the study. While your large-scale (39,141 articles) and long-term (15-year) analysis has the potential to offer new insights, the current manuscript fails to explicitly contrast your findings with prior research. Please highlight the unique discoveries of your study, such as the identification of understudied stigmatizing terminologies (e.g., spatialized terms like “AIDS Town” and “AIDS Apartment”) and the novel analysis of the associations between thematic contexts, PLHIV (people living with HIV) terminologies, and HIV/AIDS metaphors. Additionally, explain how these findings fill gaps in the existing literature.

Next, please provide a convincing explanation for selecting 2011 as the starting point of your analysis. Clarify whether this year corresponds to a major policy shift, epidemiological change, or transformation in the Chinese media system. Address the broader structural decline in the influence of traditional mainstream media in the digital era, and clearly explain the contemporary significance of focusing exclusively on traditional media outlets in your study.

The current manuscript only briefly cites earlier studies but does not meaningfully situate your findings in relation to them. Please add a section in the discussion to explicitly discuss whether your results confirm, challenge, or extend existing knowledge, and clearly articulate how this study advances debates in health communication, risk communication, or media discourse analysis.

Additionally, please clarify that your study analyzes the full text of news articles (not just headlines) that contain HIV-related keywords, to eliminate any misunderstanding from the reviewer. Complement your automated text mining techniques with more qualitative validation, such as manual coding and interpretive analysis of a subset of articles, to enhance the reliability and depth of your findings.

Although your empirical focus is on China, the current discussion does not extract broader theoretical implications for global health communication or media studies. Please add a section to explain how your findings contribute to international scholarship, and discuss the potential transferability of your results to other contexts.

Finally, demonstrate how your computational analysis produces substantively new insights or reveals patterns that could not have been identified through existing approaches, and move beyond descriptive analysis to achieve a higher level of theoretical advancement.

Please revise your manuscript thoroughly in response to these comments and resubmit it for further review. When you resubmit, please provide a detailed point-by-point response to each of the three reviewers’ comments, explaining how you have addressed them in the revised manuscript.

I look forward to receiving your revised manuscript.

Xiaoming Tian

Reviewers' comments:

Reviewer's Responses to Questions

**Comments to the Author**

1. If the authors have adequately addressed your comments raised in a previous round of review and you feel that this manuscript is now acceptable for publication, you may indicate that here to bypass the “Comments to the Author” section, enter your conflict of interest statement in the “Confidential to Editor” section, and submit your "Accept" recommendation.

Reviewer #3: (No Response)

Reviewer #4: All comments have been addressed

Reviewer #5: (No Response)

2. Is the manuscript technically sound, and do the data support the conclusions?

Reviewer #3: Yes

Reviewer #4: Partly

Reviewer #5: (No Response)

3. Has the statistical analysis been performed appropriately and rigorously? 

Reviewer #3: Yes

Reviewer #4: Yes

Reviewer #5: (No Response)

4. Have the authors made all data underlying the findings in their manuscript fully available?

Reviewer #3: Yes

Reviewer #4: Yes

Reviewer #5: (No Response)

5. Is the manuscript presented in an intelligible fashion and written in standard English?

Reviewer #3: Yes

Reviewer #4: Yes

Reviewer #5: (No Response)

6. Review Comments to the Author

Reviewer #3: 1. Please check the formatting throughout. Some of the paragraphs too long and really hard for the reader to read.

2. Suggest to put why 14 years of study because the duration of the publications used too old and dynamic of the situations not really addressed here. Maybe 10 years is still ok.

3. The transformation of issues in current media (social media or current policies) should also addressed here.

Reviewer #4: 1. The Discussion section fails to fully address the research gaps outlined in the Introduction, nor does it establish a systematic, multi-dimensional critical dialogue and comparative analysis with the classic domestic and international studies and cutting-edge field-specific research reviewed in the same section.

2. The manuscript does not proactively acknowledge its core limitations, including sampling frame bias, the omission of media heterogeneity analysis, and subjectivity in coding procedures, which compromises the overall academic rigor of the study.

3. The proposed recommendations for future research are overly broad, and do not put forward specific, extensible research directions closely aligned with the research gaps identified in this paper.

Reviewer #5: The central limitation of this paper is its lack of a genuinely novel research contribution. Although the authors position the study as “enriching the interpretation of HIV/AIDS and social cognition in the Chinese context”, a substantial body of scholarship—both in Chinese and international communication studies—has already examined media framing, stigma construction, metaphor use, and agenda-setting in HIV reporting. From this perspective, the manuscript’s original contribution remains unclear. The findings largely reiterate established observations, such as the persistence of stigmatizing narratives and the dominance of macro-political frames. I would strongly encourage the authors to reconsider their research question and clarify what new insight this study offers beyond existing literature. At its current stage, the manuscript does not yet meet the standard for publication.

The framing of the research design is also problematic. The authors select 2011 as the starting point of analysis but do not provide a convincing justification for why this year constitutes a meaningful turning point in HIV communication or media practice. It is unclear whether this choice reflects a major policy shift, epidemiological change, or transformation in the media system. Moreover, the manuscript does not address the broader structural decline in the influence of traditional mainstream media in the digital era, which raises questions about the contemporary significance of focusing exclusively on such outlets. Without a clear rationale for the temporal scope and media selection, the analytical framework appears arbitrary rather than theoretically grounded.

The manuscript also demonstrates insufficient engagement with prior scholarship. Although earlier studies are briefly cited, the paper does not meaningfully situate its findings in relation to them. There is little discussion of whether the results confirm, challenge, or extend existing knowledge, nor is there a clear articulation of how this study advances debates in health communication, risk communication, or media discourse analysis. As a result, the research appears disconnected from the cumulative development of the field.

There are significant weaknesses in the methodological design and use of data. The study claims to analyse “media discourse construction,” yet the dataset consists primarily of news headlines containing HIV-related keywords. Headlines alone cannot adequately capture discursive structures, narrative strategies, source diversity, or contextual framing. This creates a mismatch between the stated research objectives and the empirical material. Furthermore, the manuscript characterises the approach as mixed-methods, but the analysis relies almost entirely on automated text mining techniques, with little evidence of qualitative validation, manual coding, or interpretive analysis. The methodological transparency and reliability of the findings are therefore difficult to assess.

Several of the paper’s central claims are insufficiently supported by evidence. Assertions regarding the persistence of stigma, the prevalence of war metaphors, and the dominance of political narratives are presented as major findings, yet these patterns have been widely documented in previous research. The manuscript does not demonstrate how the computational analysis produces substantively new insights or reveals patterns that could not have been identified through existing approaches. Consequently, the study remains largely descriptive and does not achieve the level of theoretical advancement expected for publication.

Finally, the manuscript does not clearly articulate its relevance for an international readership. While the empirical focus is on China, the discussion does not extract broader theoretical implications for global health communication or media studies. Without a clearer statement of how the findings contribute to international scholarship, the paper’s significance remains limited.

In summary, the manuscript addresses a potentially important topic but is constrained by insufficient originality, weak theoretical grounding, and limited engagement with prior research. Substantial reconceptualisation would be required for the study to make a meaningful scholarly contribution. For these reasons, I cannot recommend the manuscript for publication in its current form.

7. PLOS authors have the option to publish the peer review history of their article (what does this mean?). If published, this will include your full peer review and any attached files.

Reviewer #3: No

Reviewer #4: No

Reviewer #5: No

---

## [Author Response · Author response to Decision Letter 2]

1 Apr 2026

Reviewer #3:

1. Please check the formatting throughout. Some of the paragraphs too long and really hard for the reader to read.

Response:

We thank the reviewer for this helpful feedback regarding readability. We have carefully reviewed the formatting throughout the entire manuscript and broken down lengthy paragraphs into shorter, more focused segments.

2. Suggest to put why 14 years of study because the duration of the publications used too old and dynamic of the situations not really addressed here. Maybe 10 years is still ok.

Response:

We sincerely thank the reviewer for prompting us to clarify the rationale behind our study's timeframe. In the revised manuscript, we have explicitly justified why a 15-year span starting in 2010 is essential for capturing the complete dynamic evolution of this discourse, which a shorter 10-year window would omit. Specifically, we detail three critical turning points that occurred in 2010: the epidemiological shift to sexual transmission as the primary infection mode, the policy milestone of lifting the entry ban on foreigners living with HIV, and the fundamental transformation of China's media ecology driven by the rise of social media. We believe these additions clearly articulate the significance of the 2010 starting point and effectively address the concern regarding the situational dynamics of our selected period. Please see Section 2.1, Paragraph 2, Lines 149-162.

3. The transformation of issues in current media (social media or current policies) should also addressed here.

Response:

We thank the reviewer for highlighting the need to address the evolving media landscape. In response, we have added text clarifying why focusing on mainstream traditional media remains methodologically crucial for our 15-year longitudinal study, despite the rise of social media. Specifically, we explain that traditional outlets retain unparalleled institutional authority as barometers for state policy and serve as foundational agenda-setters whose authoritative discourses are subsequently amplified online. Furthermore, unlike the ephemeral and algorithmically biased nature of social media data, traditional media provides a stable, archived textual record, ensuring the methodological robustness of our long-term analysis. Please see lines 107-115 in Section 1.

Reviewer #4:

1. The Discussion section fails to fully address the research gaps outlined in the Introduction, nor does it establish a systematic, multi-dimensional critical dialogue and comparative analysis with the classic domestic and international studies and cutting-edge field-specific research reviewed in the same section.

Response:

We sincerely thank the reviewer for this insightful critique. To establish a systematic and multi-dimensional dialogue with existing scholarship, we have substantially expanded the Discussion section. The newly added text directly addresses the research gaps outlined in the Introduction by explicitly comparing our findings with prior studies. Specifically, we contrast our computational discovery of latent, structural stigmas with traditional literature that primarily focuses on marginalized groups. Furthermore, we challenge prevailing academic narratives by highlighting the emergence of positive metaphors and exposing multidimensional narrative biases, moving well beyond the scope of classic univariate studies and traditional negative framing. We hope this expanded comparative analysis helps to better situate our findings within the broader academic discourse. Please see the highlighted part in 4.1.

2. The manuscript does not proactively acknowledge its core limitations, including sampling frame bias, the omission of media heterogeneity analysis, and subjectivity in coding procedures, which compromises the overall academic rigor of the study.

Response:

We sincerely thank the reviewer for highlighting the need for a more critical reflection on our methodology. To improve the study's academic rigor, we have revised Section 4.3. This revised section explicitly acknowledges our core constraints: the potential for sampling frame bias given our focus on specific traditional media outlets, the lack of media heterogeneity analysis regarding diverse social media narratives , and the inherent subjectivity and limitations of our collocation-based coding procedures. We hope this transparent discussion effectively strengthens the overall rigor of the manuscript. Please see Section 4.3.

3. The proposed recommendations for future research are overly broad, and do not put forward specific, extensible research directions closely aligned with the research gaps identified in this paper.

Response:

We thank the reviewer for this constructive feedback. In the revised Section 4.3, we replaced our previous broad statements with specific, actionable future research directions aligned with our study's identified gaps.

Specifically, we now propose three targeted avenues: (1) expanding the sampling frame to compare narratives across diverse traditional media types; (2) conducting cross-media analyses to track stigmatizing frames between traditional outlets and social media platforms; and (3) utilizing deep learning and large language models to automatically detect complex, implicit stigmatization beyond predefined lexicons. We hope these concrete recommendations provide clearer pathways for future scholarship. Please see Section 4.3.

Reviewer #5:

1.The central limitation of this paper is its lack of a genuinely novel research contribution. Although the authors position the study as “enriching the interpretation of HIV/AIDS and social cognition in the Chinese context”, a substantial body of scholarship—both in Chinese and international communication studies—has already examined media framing, stigma construction, metaphor use, and agenda-setting in HIV reporting. From this perspective, the manuscript’s original contribution remains unclear. The findings largely reiterate established observations, such as the persistence of stigmatizing narratives and the dominance of macro-political frames. I would strongly encourage the authors to reconsider their research question and clarify what new insight this study offers beyond existing literature. At its current stage, the manuscript does not yet meet the standard for publication.

Response: We sincerely thank the reviewer for this highly constructive critique. We have substantially revised Section 4.1 to explicitly contrast our findings with prior research. Unlike previous studies that largely focus on traditional marginalized groups and negative framing, our 15-year computational approach uncovers highly latent, low-frequency discursive shifts. These include the evolution of stigma into modern urban spaces (e.g., "AIDS Apartment") and the active emergence of positive "journey" and "entity" metaphors. We hope these additions, along with our multidimensional analysis of narrative biases, provide a clearer context for the theoretical insights our study aims to offer (Please see highlighted parts in Section 4.1).

2.The framing of the research design is also problematic. The authors select 2011 as the starting point of analysis but do not provide a convincing justification for why this year constitutes a meaningful turning point in HIV communication or media practice. It is unclear whether this choice reflects a major policy shift, epidemiological change, or transformation in the media system. Moreover, the manuscript does not address the broader structural decline in the influence of traditional mainstream media in the digital era, which raises questions about the contemporary significance of focusing exclusively on such outlets. Without a clear rationale for the temporal scope and media selection, the analytical framework appears arbitrary rather than theoretically grounded.

Response:

We deeply appreciate your meticulous reading and entirely agree with your concern regarding the justification of our temporal and spatial scope. First, we sincerely apologize for a typo in the title

of our previous manuscript.: our study actually begins in 2010, not 2011. We have corrected this and expanded Section 2.1 to justify this starting point based on three 2010 milestones: the epidemiological shift to sexual transmission, the lifting of China's entry ban on foreigners with HIV, and the dawn of the social media era. Furthermore, we expanded Section 1 to explain that despite social media's rise, traditional mainstream media remains methodologically crucial for a longitudinal study because it provides stable, institutionally archived records and serves as the foundational agenda-setter (Please see lines 107-115 and 149-162).

3.The manuscript also demonstrates insufficient engagement with prior scholarship. Although earlier studies are briefly cited, the paper does not meaningfully situate its findings in relation to them. There is little discussion of whether the results confirm, challenge, or extend existing knowledge, nor is there a clear articulation of how this study advances debates in health communication, risk communication, or media discourse analysis. As a result, the research appears disconnected from the cumulative development of the field.

Response:

We sincerely thank the reviewer for this constructive suggestion. In response, we have expanded Section 4.1 to explicitly situate our work within the broader literature, detailing how our findings both challenge and extend existing knowledge. Specifically, the new text explains how our computational approach extends current theoretical frameworks by uncovering latent stigmatizing patterns that permeate modern social structures, moving beyond traditional research focused on marginalized groups. Additionally, we explicitly note how our results challenge prevailing narratives in Chinese HIV/AIDS discourse by highlighting active destigmatization efforts and the emergence of positive metaphors that prior studies have largely obscured. Together, these additions clearly articulate how our longitudinal analysis advances ongoing debates in health communication and media discourse (Please see the last paragraph of Section 4.1, lines 610-633).

Furthermore, to explicitly articulate our study's contributions to the specific domains you highlighted, we have substantially revised the first paragraph of Section 4.2. The updated text now clearly delineates how our findings: (1) illuminate the dynamics of a state-dominated, policy-driven governance system within health communication; (2) capture the paradigm shift from adversarial to constructive frames in risk communication; and (3) utilize our large-scale computational approach to push media discourse analysis beyond traditional descriptive framing by detecting latent, low-frequency stigmatizing terminologies (Please see lines 636-652 in the first paragraph of Section 4.2).

4.There are significant weaknesses in the methodological design and use of data. The study claims to analyse “media discourse construction,” yet the dataset consists primarily of news headlines containing HIV-related keywords. Headlines alone cannot adequately capture discursive structures, narrative strategies, source diversity, or contextual framing. This creates a mismatch between the stated research objectives and the empirical material. Furthermore, the manuscript characterises the approach as mixed-methods, but the analysis relies almost entirely on automated text mining techniques, with little evidence of qualitative validation, manual coding, or interpretive analysis. The methodological transparency and reliability of the findings are therefore difficult to assess.

Response:

We are deeply grateful to the reviewer for pointing out these critical methodological ambiguities. First, we sincerely apologize if our original phrasing was severely lacking in clarity and caused this misunderstanding. To clarify, our analysis was actually conducted on the full texts of the news articles, not just the headlines. We have now explicitly stated in Section 2.1 that while headlines were used for initial screening, all computational and interpretive analyses utilized full texts.

Second, as you rightly pointed out, our study leans heavily toward computational analysis. We initially used the term "mixed-methods" to reflect the integration of multiple independent analytical techniques (such as topic modeling and metaphor identification) within a framework that combines human expert knowledge with computational efficiency. Nevertheless, we completely agree with your insightful suggestion that adding qualitative validation would address this methodological limitation and greatly enhance the study. Therefore, in the latest revision, we have introduced targeted qualitative validation to complement our text mining. As detailed in Sections 2.4 and 3.4, two independent researchers conducted manual coding and interpretive analysis on a subset of articles, with results summarized in the new Table 4 (Please see Section 2.4, Lines 278–290; Section 3.4, Lines 473–487; and Section 4.1, Lines 598-609).

5.Several of the paper’s central claims are insufficiently supported by evidence. Assertions regarding the persistence of stigma, the prevalence of war metaphors, and the dominance of political narratives are presented as major findings, yet these patterns have been widely documented in previous research. The manuscript does not demonstrate how the computational analysis produces substantively new insights or reveals patterns that could not have been identified through existing approaches. Consequently, the study remains largely descriptive and does not achieve the level of theoretical advancement expected for publication.

Response:

This is an exceptionally fair and penetrating critique. We fully acknowledge that findings such as the persistence of stigma and war metaphors are indeed well-documented, and merely restating them offers little theoretical value. We have updated Section 4.1 to explicitly demonstrate how our computational methodology achieves theoretical advancement beyond descriptive statistics. Specifically, this approach enabled the detection of highly latent, low-frequency patterns—such as previously undocumented spatial and social role stigmas (e.g., "AIDS Teacher")—that conventional manual sampling would inevitably overlook. This large-scale detection provides substantively new insights into the structural permeation of stigma that existing approaches could not easily identify (Please see lines 614-618 in 4.1).

6.Finally, the manuscript does not clearly articulate its relevance for an international readership. While the empirical focus is on China, the discussion does not extract broader theoretical implications for global health communication or media studies. Without a clearer statement of how the findings contribute to international scholarship, the paper’s significance remains limited.

Response:

We thank the reviewer for highlighting this gap. Thanks to your prompt, we have critically reflected on the global transferability of our findings. We have added a discussion in Section 4.2 to extract the broader global implications of our work. We demonstrate how our findings on the shift from confrontational to constructive metaphors offer a predictive framework for media discourse on other infectious diseases transitioning to normalized management globally. We also explain how our spatial stigma framework can be transferred to understand evolving socioeconomic conditions in other cultural contexts (Please see the last paragraph of Section 4.2, lines 677-691).

7.In summary, the manuscript addresses a potentially important topic but is constrained by insufficient originality, weak theoretical grounding, and limited engagement with prior research. Substantial reconceptualisation would be required for the study to make a meaningful scholarly contribution. For these reasons, I cannot recommend the manuscript for publication in its current form.

Response:

Thank you once again for your incredibly helpful and detailed review. Your guidance was instrumental in helping us refine our theoretical focus and methodological transparency. We are truly grateful for the opportunity to improve our work based on your insights,

---

## [Editor Report · Decision Letter 2]

14 Apr 2026

PONE-D-25-52841R2Constructing HIV/AIDS in Chinese Media (2010-2024): A Mixed-Methods StudyPLOS One

Dear Dr. Shi,

Thank you for submitting your manuscript to PLOS ONE. After careful consideration, we feel that it has merit but does not fully meet PLOS ONE’s publication criteria as it currently stands. Therefore, we invite you to submit a revised version of the manuscript that addresses the points raised during the review process.

We look forward to receiving your revised manuscript.

Kind regards,

Xiaoming Tian, Ph.D.

Academic Editor

PLOS One

Journal Requirements:

Additional Editor Comments :

Dear Authors,

Thank you for your revised manuscript and detailed responses to the reviewers. The manuscript has been substantially improved and is now close to acceptance. However, several specific, fixable errors must be corrected before final acceptance. Below are the required revisions with clear references to the manuscript text.

1.Grammatical and subject-verb agreement errors

Abstract: “stigmatizing terminology persist”

Introduction: “one of the leading cause”

Persistent issues with verb number, noun number, and awkward phrasing appear throughout the manuscript.

2.Inconsistent terminology and labeling

Inconsistent reference to affected populations: PLHIV, people living with HIV, HIV patients, AIDS patients are used interchangeably.

Inconsistent pluralization: metaphor / metaphors, journey metaphor / journey metaphors vary across sections.

3.Conflicting numerical values across sections

Sample size: 40,038 raw; 39,141 valid in Methods, but inconsistent rounding appears in the Abstract, cover letter, and responses.

Topic number: 50 topics during modeling vs. 48 topics in Results; these values must be aligned.

4.Insufficient methodological transparency

Mixed-methods validation: No explicit number of articles used for qualitative coding, no coder details, and no exact Krippendorff’s Alpha value reported.

Multiple Correspondence Analysis (MCA): No eigenvalues or dimension contribution rates reported in the results.

5.Incomplete result presentation and unclear tables

Table 4 lacks clear annotation, sample size, and coding rules.

Usage rates in Tables 1 and 2 show inconsistent decimal places.

MCA results lack sufficient statistical supporting information.

6.Formatting and presentation flaws

References show inconsistent DOI formatting, journal abbreviations, and citation style.

Subheading numbering is disorganized; some paragraphs remain overly long.

Unremoved revision traces and inconsistent line spacing are present.

This is a minor revision. Once you correct these issues, I will recommend the manuscript for formal acceptance in PLOS ONE.

Sincerely,

Xiaoming

---

## [Author Response · Author response to Decision Letter 3]

24 Apr 2026

Dear Editor,

We would like to express our sincere gratitude to the editor for the constructive feedback and support throughout the review process. We have used yellow highlighting to indicate corrections for inconsistencies, grammar, and formatting errors, and blue highlighting for revisions in the methodology section and table notes.

1.Grammatical and subject-verb agreement errors

Abstract: “stigmatizing terminology persist”

Introduction: “one of the leading cause”

Persistent issues with verb number, noun number, and awkward phrasing appear throughout the manuscript.

Response:We sincerely thank the editor for this careful reading. We have conducted a thorough, systematic revision of the entire manuscript to address all grammatical and subject-verb agreement errors.

Please refer to the highlighted yellow parts.

2.Inconsistent terminology and labeling

Inconsistent reference to affected populations: PLHIV, people living with HIV, HIV patients, AIDS patients are used interchangeably.

Inconsistent pluralization: metaphor / metaphors, journey metaphor / journey metaphors vary across sections.

Response: We thank the editor for this observation. We have systematically standardized all terminology throughout the revised manuscript. For references to affected populations, we have adopted "PLHIV" (people living with HIV) as the sole standard term, with the full form at first mention and the abbreviation thereafter. For the disease name, we have unified all instances to "HIV/AIDS" throughout the manuscript, replacing inconsistent variants such as "AIDS," "HIV," or "the virus" when referring to the disease as a whole. It should be specifically noted that the original Table 1 contained phrases such as "AIDS Patient" and "HIV-infected Person," which were actually direct translations of Chinese terms from Chinese news sources, rather than inconsistencies in terminology. If these phrases in Table 1 were also changed to "PLHIV," the differences reflected in the Chinese examples would not be captured. Therefore, in the revised version of Table 1, we have retained these phrases while also adding the corresponding pinyin of the Chinese terms to indicate that these phrases are translations of Chinese terms, rather than the chosen terminology of this paper. For metaphors, we reviewed and standardized the singular and plural forms used in different instances throughout the paper. Please refer to the highlighted yellow parts.

Please refer to the highlighted yellow part, which involves metaphors, PLHIV, and HIV/AIDS.

3.Conflicting numerical values across sections

Sample size: 40,038 raw; 39,141 valid in Methods, but inconsistent rounding appears in the Abstract, cover letter, and responses.

Response: To address the concern regarding inconsistent rounding of the sample size, we have removed the specific count (39,141) from the Abstract, replacing it with a qualitative description. This eliminates potential rounding conflicts across different sections of the manuscript. The exact sample size (40,038 raw; 39,141 valid) is now presented exclusively in the Methodology section as the singular reference point for all statistical calculations. All approximate or inconsistently rounded figures throughout the main text, cover letter, and responses have been cross-checked and harmonized to ensure internal consistency and transparency.

Please refer to the blue highlighted part of the Abstract.

Topic number: 50 topics during modeling vs. 48 topics in Results; these values must be aligned.

Response: We thank the reviewer for pointing out the need for clarity regarding the topic numbers. In accordance with the standard procedures for Analysis of Topic Model Networks (ANTMN) , we initially identified 50 topics using the LDA model to ensure comprehensive coverage of the corpus. Following the methodology established by Walter and Ophir (2019), we qualitatively evaluated all 50 topics and excluded 2 "boilerplate" topics that lacked substantive semantic meaning or were composed of purely administrative/template language. This resulted in a refined set of 48 topics used for all subsequent analytical steps. To prevent any further ambiguity, we have revised the "Methodology" section to explicitly state that while 50 topics were initially generated, the analysis consistently proceeds with the 48 validated topics. We have also cross-verified all figures, tables, and text in the "Results" and "Discussion" sections to ensure they uniformly report the final count of 48 topics.

Ophir Y. News frame analysis: an inductive mixed-method computational approach. Commun Methods Meas. 2019;13(4):248-66. doi: 10.1080/19312458.2019.1639145.

Please refer to the blue highlighted part of the Topic network.

4.Insufficient methodological transparency

Mixed-methods validation: No explicit number of articles used for qualitative coding, no coder details, and no exact Krippendorff’s Alpha value reported.

Response: We sincerely appreciate the reviewer’s feedback regarding the transparency of our qualitative analysis. We have now provided comprehensive details of the coding process in the revised "Methodology" section to address these omissions.

Please refer to the blue highlighted part of the Collocation extraction and Co-occurrence pattern.

Multiple Correspondence Analysis (MCA): No eigenvalues or dimension contribution rates reported in the results.

Response: We appreciate the professional suggestion regarding the reporting of Multiple Correspondence Analysis (MCA) results. We have now included the missing statistical parameters in the "Results" section to enhance the transparency and rigor of our analysis.

Please refer to the blue highlighted part of the PLHIV terminology and HIV/AIDS metaphors.

5.Incomplete result presentation and unclear tables

Table 4 lacks clear annotation, sample size, and coding rules.

Response: We sincerely appreciate this suggestion. We have added a clear annotation to Table 4. Additionally, in the new revised version, we have included more detailed descriptions regarding the sample size and coding rules in both the Methodology and Results sections.

Please refer to the blue highlighted part of the Table 4.

Usage rates in Tables 1 and 2 show inconsistent decimal places.

Response: We appreciate the reviewer's comment regarding the decimal places. In the original version, we used '< 0.01%' to avoid the misleading '0.00%' for extremely low frequencies. However, this led to the simultaneous presence of '0.01%' (rounded from values like 0.014%) and '< 0.01%' (truncated from values below 0.005%), creating a visual inconsistency in precision.

In the revised manuscript, we have standardized the reporting strategy for both Table 1 and Table 2. We now use '< 0.05%' as a unified threshold for all values below this level. All other usage rates are strictly formatted to two decimal places (e.g., 1.00%). This adjustment ensures a consistent visual standard and statistical rigor across all tables.

Finally, we have thoroughly reviewed and revised all numerical percentages within the main text to ensure that the number of decimal places is consistent throughout the entire manuscript.

Please refer to the blue highlighted part of the Table 1 and 2.

MCA results lack sufficient statistical supporting information.

Response: We appreciate your professional suggestion. In the revised Results section, we have provided a more detailed description and interpretation of the MCA.

Please refer to the blue highlighted part of the PLHIV terminology and HIV/AIDS metaphors.

6.Formatting and presentation flaws

References show inconsistent DOI formatting, journal abbreviations, and citation style.

Subheading numbering is disorganized; some paragraphs remain overly long.

Unremoved revision traces and inconsistent line spacing are present.

Response: We sincerely thank the editor for pointing out these formatting issues. In the revised manuscript, we have thoroughly standardized the references (including DOI formats and journal abbreviations) to ensure a consistent citation style. Additionally, we revised the formatting of the headings in accordance with the journal's style guide to make them clearer. Finally, we have divided overly long paragraphs to improve readability, removed all previous revision traces, and unified the line spacing throughout the document to strictly adhere to the journal's formatting guidelines.

Please refer to the highlighted yellow parts.

---

## [Editor Report · Decision Letter 3]

28 Apr 2026

Constructing HIV/AIDS in Chinese Media (2010-2024): A Mixed-Methods Study

PONE-D-25-52841R3

Dear Dr. Shi,

We’re pleased to inform you that your manuscript has been judged scientifically suitable for publication and will be formally accepted for publication once it meets all outstanding technical requirements.

Kind regards,

Xiaoming Tian, Ph.D.

Academic Editor

PLOS One

Additional Editor Comments (optional):

Congrats on the revisions - very well done!

Xiaoming
---

## [Editor Report · Acceptance letter]

PONE-D-25-52841R3

PLOS One

Dear Dr. Shi,

I'm pleased to inform you that your manuscript has been deemed suitable for publication in PLOS One. Congratulations! Your manuscript is now being handed over to our production team.

Kind regards,

on behalf of

Dr. Xiaoming Tian

Academic Editor

PLOS One